# DIFFERENTIABLE OPTIMIZATION IN PLANE-WAVE DENSITY FUNCTIONAL THEORY FOR SOLID STATES

## ABSTRACT

Plane-wave density functional theory is a computational quantum mechanical modeling method used to investigate the electronic structure of solids. It employs plane-waves as the basis set for representing electronic wave functions and leverages density functional theory to compute the electronic structure properties of many-body systems. Traditionally, the Self-Consistent Field (SCF) method is predominantly adopted for optimization in current DFT computations. However, this method encounters notable convergence and computational challenges, and its iterative nature obstructs the incorporation of emergent deep learning enhancements. To address these challenges, we introduce a fully differentiable optimization method tailored to resolve the intrinsic challenges associated with the optimization of plane-wave density functional methods. This methodology includes a direct total energy minimization approach for solving Kohn-Sham equations in periodic crystalline systems, which is coherent with deep learning infrastructures. The efficacy of our approach is illustrated through its two applications in solid-state physics: electron band structure prediction and geometry optimization. Our enhancements potentially pave the way for various gradient-based applications within deep learning paradigms in solid-state physics, extending the boundaries of material innovation and design. We illustrate the utility and diverse applications of our method on real crystal structures and compare its effectiveness with several established SCF-based packages, demonstrating its accuracy and robust convergence property.

## 1 INTRODUCTION

Kohn-Sham density functional theory (KS-DFT) (Kohn & Sham, 1965) has become the primary tool for quantum mechanical electronic structure analysis in solid-state physics and materials science, garnering substantial interest in both academic and industrial settings in recent decades (Pribram-Jones et al., 2015). Analyses of the usage of supercomputing resources substantiate that KS-DFT often constitutes a predominant consumer of computational time within numerous research facilities (Austin et al., 2020). Within the framework of KS-DFT, plane-wave methods enable researchers to explore and predict diverse solid-state material properties, including magnetism, superconductivity, and thermal conductivity, with satisfying accuracy. These approaches facilitate a thorough understanding of the various attributes of materials, spurring advancements in the development of innovative materials and their applications in fields such as superconductors (Oliveira et al., 1988), energy storage (Spotte-Smith et al., 2022), and nanotechnology (Frink et al., 2002), among many others.

The Self-Consistent Field (SCF) method is widely used in solving density functional theory methods. It typically commences by initializing an electron density—usually a superposition of atomic densities—from which an effective potential is deduced. This deduced potential is iteratively leveraged to solve the Kohn-Sham equations, resulting in a set of orbitals that are then used to calculate a new electron density. This iterative process continues until the electron density converges within a set tolerance, thus determining the system's ground state energy through the attainment of self-consistency. However, this methodology is computationally demanding, especially for systems characterized by numerous electrons or complex electron interactions, with the attainment of convergence presenting notable challenges and often requiring the implementation of advanced mixing or damping strategies, or failing to find a stable solution altogether (Yang et al., 2007; Lehtola et al., 2020; Cancès et al., 2021; Schlegel & McDouall, 1991). Furthermore, the optimization of other system parameters, such as atom geometry and lattice, as well as direct optimization with respect to crystal property metrics,

poses additional challenges in SCF methods. These challenges are attributable to the intricate implementation routines and the requirement to address eigendecomposition problems in every iteration. As a result, conventional practices necessitate the repetitive execution of SCF calculations following each update, which is a procedure that is not only resource-intensive but also susceptible to numerical instability (Yang et al., 2009).

With the advancement of deep learning, a multitude of innovative optimization methods and infrastructures have emerged in the past years, providing diverse and efficient solutions for solving various models and algorithms. To solve the Kohn-Sham equations, the iterative nature of the prevalent SCF optimization method, extensively applied in computational physics and materials science, presents significant challenges to enhance the DFT methods with deep learning techniques (e.g. requiring back propagation of a derivative method through an SCF loop that must reach convergence without failure, as in (Kasim & Vinko, 2021)). In this paper, we propose a fully differentiable optimization approach, specifically designed to overcome the inherent challenges associated with solving plane-wave density functional methods, allowing for a more harmonious integration with contemporary deep learning techniques and infrastructures.

In this study, we introduce a novel direct optimization strategy, aiming to circumvent the necessity of solving the conventional SCF loop within the plane-wave DFT framework. We illustrate this approach through its application to differentiable and direct total energy optimization and demonstrate a direct optimization method for attaining the Kohn-Sham eigenvalues. These eigenvalues construct the electron band structure, unveiling the intricate electronic properties of materials. Further, we develop a gradient-based algorithm for geometry optimization, which is also fully differentiable. To validate the effectiveness of our proposed methodology, we implement our approach with deep learning framework JAX (Bradbury et al., 2018), and conduct experiments on realistic crystals and draw comparisons with existing implementations of density functional theory in solid states. Additionally, we furnish a detailed convergence analysis, contrasting gradient-based optimizers with the SCF approach, to underscore the comparative merits of our methodology.

The main **contributions** of this study are as follows:

- We present a fully differentiable approach for direct total energy minimization to solve the Kohn-Sham equations in periodic solid-state systems. This approach makes it possible to leverage recent developments in deep learning, including auto-differentiability and a plethora of well-established gradient-based optimizers.

- We put forth a direct optimization technique in two important applications in material science, the electron band structure prediction and geometry optimization. Experiments on realistic systems demonstrate the effectiveness of our approach.

- This research presents the potential for bridging the existing divide between the emerging deep-learning infrastructures and scientific computing approach in the realms of solid-state physics and material science.

## 2  DIFFERENTIABLE OPTIMIZATION OF TOTAL ENERGY IN KOHN-SHAM DFT

Density Functional Theory (DFT) is a quantum mechanical formalism based on the fact that ground-state electronic density determines a system's wave function and energy, as per the Hohenberg-Kohn theorem (Hohenberg & Kohn, 1964). Being able to solve electronic structures with only density make it cheaper than wavefunction theories makes it a key tool for studying electronic structures.

**Kohn-Sham DFT**   However, since the exact form of the universal density functional in HK formalism is not known, most practical DFT applications relies on the Kohn-Sham formalism (Kohn & Sham, 1965), which is a mean-field wavefunction theory that uses the single Slater determinant formed by orthonormal single-particle wave functions $\{\psi_i(\boldsymbol{r})\}_{i=1}^I$ as ansatz. The electronic density is then given by $\rho(\boldsymbol{r}) = \sum_{i=1}^I |\psi_i(\boldsymbol{r})|^2$, and the ground state density can then be obtained by minimizing the total electronic energy subject to the orthonormal constraint:

$$
\begin{aligned}
\min_{\{\psi_i\}} \quad & E_{\text{el}}[\{\psi_i\}] = E_{\text{kin}}[\{\psi_i\}] + E_{\text{ext}}[\rho] + E_{\text{har}}[\rho] + E_{\text{xc}}[\rho], \\
\text{s.t.} \quad & \langle \psi_i | \psi_j \rangle = \delta_{ij}.
\end{aligned}
\tag{1}
$$

where $E_{\text{kin}}$, $E_{\text{ext}}$, $E_{\text{har}}$, $E_{\text{xc}}$ denote the kinetic energy, external potential energy, Hartree energy, and the exchange-correlation energy. This constraint optimization problem can be solved via the Lagrange multiplier method, and the first order condition of the Lagrangian gives the Kohn-Sham equation

$$\hat{H}^{\text{KS}}[\rho]\psi_i = \varepsilon_i \psi_i \tag{2}$$

where $\hat{H}^{\text{KS}}[\rho]$ is the KS Hamiltonian that depends on the electronic density $\rho(\boldsymbol{r})$, and $\varepsilon_i$ are the KS eigenvalues. The KS equation is usually solved via SCF. Further details are provided in Appendix D.

**Periodic crystalline system** Crystalline systems are usually simulated by solving Schrodinger's equation in a Bravais lattice, which captures the periodic structure of the material by tiling a parallelepiped (unit cell) defined by lattice constants $\boldsymbol{a}_i \in \mathbb{R}^3, i \in \{1, 2, 3\}$ in space. The lattice is subjected to periodic boundary condition (PBC), which means the lattice itself is infinitely tiled in space. We denote the number of unit cell in the lattice as $M = M_1 \times M_2 \times M_3$, and each cell in the lattice can be indexed by $\boldsymbol{m} := (m_1, m_2, m_3)$ where $m_i = 0, 1, \cdots, M_i - 1$ for $i \in \{1, 2, 3\}$. The external potential is periodic in the lattice , so by Bloch's theorem (Bloch, 1929), the electronic wavefunction only differs by a phase shift between different cells, i.e.

$$\psi_{\boldsymbol{m}}(\boldsymbol{r}) = \exp\left(\mathrm{i}\boldsymbol{k}_{\boldsymbol{m}}^\top \boldsymbol{r_n}\right) u_{\boldsymbol{m}}(\boldsymbol{r}), \quad \boldsymbol{k_m} = \sum_{i=1}^{3} \frac{m_i}{M_i} \boldsymbol{b}_i \tag{3}$$

where $u_{\boldsymbol{m}}$ is a function periodic in the lattice, $\boldsymbol{b}_i$ are the reciprocal lattice vector. The wavevectors $\boldsymbol{k_m}$ takes discrete values due to the PBC.

**DFT in periodic systems** If KS formalism is applied to a periodic system, the electronic density will be distributed among different wavevector $\boldsymbol{k_m}$:

$$\rho(\boldsymbol{r}) = \sum_{\boldsymbol{m}} \sum_{i=1}^{I} |\psi_{i,\boldsymbol{m}}(\boldsymbol{r})|^2. \tag{4}$$

Eq. 1 and 2 still holds and the only difference is that we now have $I \times M$ orbitals $\{\psi_{i,\boldsymbol{m}}\}$. By Eq. 3, the orthonormal constraint simplifies to $\langle \psi_{i,\boldsymbol{m}} | \psi_{j,\boldsymbol{m}'} \rangle = \delta_{i,j}$ since orbitals with different wavevector are automatically orthogonal, and the KS equation decouples into $M$ equations

$$\hat{H}_{\boldsymbol{m}}^{\text{KS}}[\rho] u_{i,\boldsymbol{m}}(\boldsymbol{r}) = u_{i,\boldsymbol{m}}(\boldsymbol{r}) \varepsilon_{i,\boldsymbol{m}} \tag{5}$$

which can be solved separately for each $\boldsymbol{k_m}$. The KS eigenvalues $\varepsilon_{i,\boldsymbol{m}}$ here determines the band structure of the material.

**Direct optimization in plane-wave basis** There are many ways to parameterize $u_{i,\boldsymbol{m}}(\boldsymbol{r})$, and the most commonly used method is via Fourier basis. Define the mesh of the Fourier basis with size $N = N_1 \times N_2 \times N_3$ within the confines of the unit cell, then at each lattice point $\boldsymbol{r_n} = \sum_{i=1}^{3} \frac{n_i}{N_i} \boldsymbol{a}_i$ where $\boldsymbol{n} := (n_1, n_2, n_3)$ and $n_i = 0, 1, \ldots, N_i - 1$, the value of $u_{i,\boldsymbol{m}}(\boldsymbol{r_n})$ can be represented by a linear combination of plane-waves

$$u_{i,\boldsymbol{m}}(\boldsymbol{r_n}) = \sum_{\boldsymbol{n}'} c_{i,\boldsymbol{m},\boldsymbol{n}'} \exp\left(\mathrm{i}\boldsymbol{G}_{\boldsymbol{n}'}^\top \boldsymbol{r_n}\right) \tag{6}$$

where $\boldsymbol{G_n} = \sum_{i=1}^{3} n_i \boldsymbol{b}_i$ lies on the reciprocal FFT-mesh, and $c_{i,\boldsymbol{m},\boldsymbol{n}'}$ are the Fourier coefficients of $u_{i,\boldsymbol{m}}$. With this parameterization, we can use the technique proposed in (Li et al., 2023) to perform direct optimization of Eq. 1. Specifically, to enforce the constraint $\langle \psi_{i,\boldsymbol{m}} | \psi_{j,\boldsymbol{m}'} \rangle = \delta_{i,j}$, for each $\boldsymbol{m}$ we have

$$\boldsymbol{C_m} := [c_{i,\boldsymbol{m},\boldsymbol{n}}]_{i,\boldsymbol{n}} = \mathrm{QR}\left(\boldsymbol{W_m}\right) \in \mathbb{C}^{I \times N} \tag{7}$$

where $\mathrm{QR}$ is a function that returns the orthogonal part from a QR decomposition, $\boldsymbol{W_m} \in \mathbb{C}^{I \times N}$ is the learnable parameter. The integrals in the energy functional $E_{\text{el}}[\{\psi_{i,\boldsymbol{m}}\}]$ (Eq. 1) has analytical formulae in Fourier space, and the detailed derivation can be found in the appendix.

**Computational graph** A differentiable computational graph of our method is presented in Fig. 1.

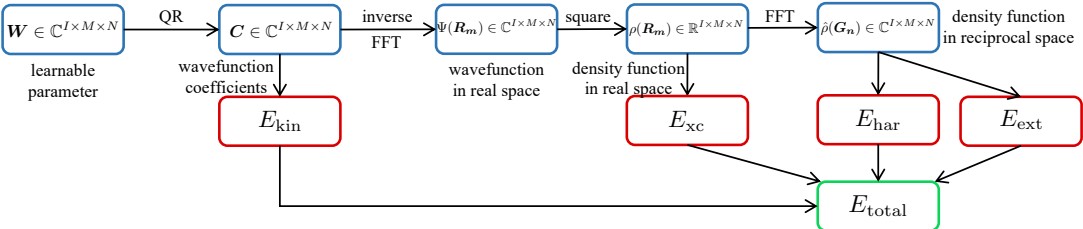

Figure 1: The computational graph shows the methodology for total energy minimization. Within this graph, arrows symbolize the sequence of forward computations. Every operation within this computational graph is differentiable, facilitating the gradient backpropagation.

## 3 Direct Optimization of Electronic Band Structure

The electronic band structure represents a cornerstone in the domain of solid-state physics, instrumental in delineating the electronic behavior within crystalline solids. Serving as a holistic representation, the band structure imparts pivotal information regarding a material's intrinsic properties. It describes whether a material exhibits conductive, insulating, or semiconducting behavior. Additionally, it provides insights into the material's optical properties, magnetic tendencies, and thermal conductivity attributes, thereby offering a comprehensive understanding vital for diverse technological applications. This section introduces a fully differentiable direct optimization method tailored for the computation of the electronic band structure.

**Method** For a fixed k-point $\boldsymbol{k_m}$, the KS Hamiltonian in Eq. 5 represented in the Fourier basis $|\boldsymbol{k_m} + \boldsymbol{G_n}\rangle = \exp\left(\mathrm{i}(\boldsymbol{k_m} + \boldsymbol{G_n})^\top \boldsymbol{r}\right)$ is a $N \times N$ matrix $\boldsymbol{F_m}$, where the matrix element is given by

$$\boldsymbol{F_{m,n,n'}}[\rho^*] = \langle \boldsymbol{k_m} + \boldsymbol{G_n}|\hat{H}_m^{\mathrm{KS}}[\rho^*]|\boldsymbol{k_m} + \boldsymbol{G'_n}\rangle, \tag{8}$$

Using the Fourier coefficient $\boldsymbol{C_m} \in \mathbb{C}^{I \times N}$, we can write the KS equation in matrix form

$$\boldsymbol{F_m}[\rho^*]\boldsymbol{C_m^\dagger} = \boldsymbol{C_m^\dagger}\varepsilon_{i,m}, \tag{9}$$

which require us to diagonalize a $N \times N$ matrix for each sampled $\boldsymbol{k_m}$. However, we are only interested in the lowest $I$ eigenvalues of this matrix. To extract these, we simply minimize the sum of the eigenvalues of the smaller $I \times I$ matrix $\boldsymbol{C_m}\boldsymbol{F_m}[\rho^*]\boldsymbol{C_m^\dagger}$, i.e., calculate

$$\min_{\boldsymbol{W_m} \in \mathbb{C}^{I \times N}} \mathrm{tr}\left(\boldsymbol{C_m}\boldsymbol{F_m}[\rho^*]\boldsymbol{C_m^\dagger}\right) \tag{10}$$

where the coefficients $\boldsymbol{C_m}$ are obtained from the QR decomposition of $\boldsymbol{W_m}$ as described in Eq. 7. There are two main advantages of this approach: (1) The orthogonality constraint is seamlessly converted to an unconstrained form using the QR reparameterization trick, which makes the band structure optimization problem a direct optimization process that aligns well with the deep learning paradigm; (2) Typically, $I \ll N$, so by only storing the $I \times I$ matrix $\boldsymbol{C_m}\boldsymbol{F_m}[\rho^*]\boldsymbol{C_m^\dagger}$ we drastically reduce the memory requirement.

In summary, our method is comprised of the following steps:

- Select $\boldsymbol{k}$-mesh size $M_1 \times M_2 \times M_3$, then run the direct optimization process to obtain the ground state electron density $\rho^*$, as described in section 2.
- For each $\boldsymbol{k_m}$ in the $\boldsymbol{k}$-path, use the ground state density $\rho^*$ to construct the matrix representation of the KS Hamiltonian $\boldsymbol{F_m}[\rho^*]$. Then obtain coefficients $\boldsymbol{C_m}$ by solving Eq. 10.
- Perform a one-step diagonalization of $\boldsymbol{C_m}\boldsymbol{F_m}[\rho^*]\boldsymbol{C_m^\dagger}$ to obtain KS eigenvalues $\varepsilon_{i,m}$, which represent the electronic band structure.

**Fine-tuning along the $\boldsymbol{k}$ path** Although in the above algorithm, the calculation for each $\boldsymbol{k_m}$ can be parallelized, when running on hardware with limited memory we have to process each $\boldsymbol{k_m}$ sequentially. Instead of randomly initializing the parameters $\boldsymbol{W_m}$ for each $\boldsymbol{k_m}$, we use the parameter

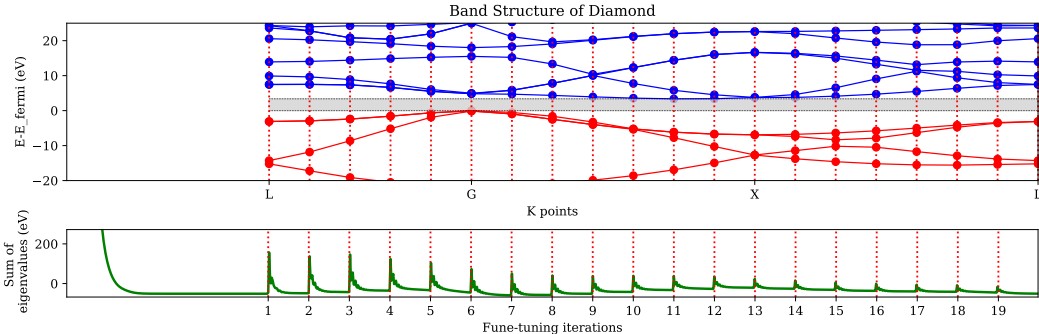

Figure 2: An illustration of the fine-tuning process for band structure calculation on the diamond structure using LDA functionals. Top: the band structure optimized via our direct optimization method is shown. The valence and conduction bands are represented by red and blue lines, respectively. A distinct band gap is highlighted in the grey area. Bottom: we present the training curve corresponding to the minimization process. For optimization, we employed the Adam optimizer (Kingma & Ba, 2015). We use 4000 iterations for the first $\boldsymbol{k}$ point and 100 for fine-tuning the subsequent $\boldsymbol{k}$ points.

$\boldsymbol{W_{m'}}$ obtained from the previous step as an initial guess and fine-tune it to determine the parameters for the subsequent $\boldsymbol{k}$-point. Given the continuity of the band structure, only a few fine-tuning iterations are needed, as shown in Figure 2. This method significantly cuts down computation time without demanding additional computational resources.

## 4 DIFFERENTIABLE GEOMETRY OPTIMIZATION

Differentiability is especially useful for jointly optimizing other parameters in the system as in the task of geometry optimization. The problem of geometry optimization is to find the equilibrium geometry, which is atom configuration $\boldsymbol{R} = \{\boldsymbol{\tau}_\ell\}$ that minimizes the total electronic and ionic energy of the system $E_{\text{tot}}(\boldsymbol{C}, \boldsymbol{R}) = E_{\text{el}}(\boldsymbol{C}, \boldsymbol{R}) + E_{\text{nuc}}(\boldsymbol{R})$, where $\boldsymbol{C}$ is the basis coefficients of the electronic wavefunction and is subjected to the orthonormal constraint, and $E_{\text{nuc}}$ is the nuclear repulsion energy

$$E_{\text{nuc}}(\boldsymbol{R}) = \frac{1}{2} \sum_{\ell}^{L} \sum_{\ell'}^{L} \sideset{}{'}\sum_{\boldsymbol{m}} \frac{q_\ell q_{\ell'}}{|\boldsymbol{\tau}_\ell - \boldsymbol{\tau}_{\ell'} + \boldsymbol{R_m}|}, \tag{11}$$

where $\boldsymbol{\tau}_\ell$ are atomic positions, $q_\ell$ are the corresponding charge distributions, indexed by $\ell = 1, 2, \ldots, L$, where $L$ represents the total number of atoms within the unit-cell, and $\sum'_{\boldsymbol{m}}$ allows consideration of interactions beyond the primitive cell where $\boldsymbol{m} = (m_1, m_2, m_3)$ represents an integer translation of the lattice vectors, we exclude the terms where $\ell = \ell'$. Geometry optimization is an important problem since equilibrium geometry provides a fundamental understanding of a material. An accurate atomic structure, obtained through optimization, is essential to precisely compute electronic structure, and other properties of the material like mechanical, thermal, and vibrational properties.

From the problem definition, the most natural way to solve geometry optimization is to perform constraint minimization of the total energy $E_{\text{tot}}(\boldsymbol{C}, \boldsymbol{R})$ by varying the wavefunction $\boldsymbol{C}$ and geometry $\boldsymbol{R}$ at the same time, i.e.

$$\min_{\boldsymbol{C}, \boldsymbol{R}} \quad E_{\text{tot}}(\boldsymbol{C}, \boldsymbol{R}),$$
$$\text{s.t.} \quad \boldsymbol{C_m^\dagger} \boldsymbol{C_m} = \boldsymbol{I}. \tag{12}$$

However, when using the SCF approach, to ensure convergence the update in $\boldsymbol{C}$ is usually dampened, i.e. history-dependent, and interleaving geometry update will disrupt the convergence of SCF. Therefore when using SCF, the force method (Pulay, 1969) is usually used, which is usually given by the Hellman-Feynman theorem (Feynman, 1939) (derivation in Appendix H.1)

$$\boldsymbol{F}_{\text{HF}} = -\frac{\mathrm{d}\tilde{E}_{\text{tot}}(\boldsymbol{R})}{\mathrm{d}\boldsymbol{R}} = -\int \mathrm{d}\boldsymbol{r} \, \tilde{\rho}(\boldsymbol{r}) \frac{\partial V_{\text{ext}}(\boldsymbol{r}; \boldsymbol{R})}{\partial \boldsymbol{R}} - \frac{\partial E_{\text{nuc}}(\boldsymbol{R})}{\partial \boldsymbol{R}}. \tag{13}$$

The comparisons between the two methods are outlined in Algorithms 1 and 2.

| **Algorithm 1** SCF Geometry Optimization | **Algorithm 2** Direct Geometry Optimization |
|---|---|
| **Input:** learning rate $\omega$ | **Input:** learning rate $\omega$ |
| 1: Initialize geometry $\boldsymbol{R}$ and parameter $\boldsymbol{C}$; | 1: Initialize geometry $\boldsymbol{R}$ and parameter $\boldsymbol{C}$; |
| 2: **while** not converged **do** | 2: **while** not converged **do** |
| 3:    update $\boldsymbol{C}$ by performing SCF; | 3:    update $\boldsymbol{C} \leftarrow \boldsymbol{C} + \omega \cdot \partial E_{\text{tot}}/\partial \boldsymbol{C}$ ; |
| 4:    update $\boldsymbol{R} \leftarrow \boldsymbol{R} + \omega \boldsymbol{F}_{\text{HF}}(\boldsymbol{R}, \boldsymbol{C})$; | 4:    update $\boldsymbol{R} \leftarrow \boldsymbol{R} + \omega \cdot \partial E_{\text{tot}}/\partial \boldsymbol{R}$; |
| 5: **end while** | 5: **end while** |
| 6: return $\boldsymbol{R}$ | 6: return $\boldsymbol{R}$ |

Our approach is fully differentiable, allowing us to directly perform joint optimization of Eq. 12 utilizing gradient descent. It is important to note that the nuclear repulsion indicated in Eq. 11 is not straightforwardly evaluable in real space due to its slow convergence. Consequently, the employment of a technique known as Ewald summation (Ewald, 1921) is needed. For the Ewald summation, we follow the derivation in (Tsili et al., 2023) and (Baroni et al., 2001). The detailed equations are provided in Appendix H.2.

## 5 EXPERIMENTS

In this section, we assess our methodology through experiments conducted on real-world crystals and materials. Our evaluation is primarily centered on three subtasks: electronic band structure prediction, geometry optimization and convergence analysis.

We implement our approach with the deep learning framework JAX (Bradbury et al., 2018). All the experiments of our approach are conducted on an NVIDIA A100 GPU with 40GB memory. All remaining experiments with other implementations are conducted on a server powered by an Intel Xeon CPU @ 2.10GHz and furnished with 64GB of memory.

### 5.1 ELECTRON BAND STRUCTURE PREDICTION

In this section, we assess the accuracy and efficacy of our direct optimization method by comparing it to existing implementations in the context of electron band structure prediction, which is a crucial step for uncovering the electronic properties of materials. We conduct tests on four distinct crystal structures: lithium (Wyckoff, 1963), aluminum (Mulder et al., 2010), with crystal structures obtained from the Open Crystallographic Database (OCD) (Gražulis et al., 2012), and carbon (diamond), and silicon, with crystal structures obtained from the critic2 (Otero-de-la Roza et al., 2014) library.

**Baselines** We conduct comparisons with several packages, including Quantum ESPRESSO (QE) (Giannozzi et al., 2009), Fritz Haber Institute *ab initio* materials simulation (FHIaims) package (Blum et al., 2009), and GPAW (Enkovaara et al., 2010). QE conducts DFT calculations using plane-wave basis sets and pseudopotentials to represent electron-ion interactions. As we use a plane-wave only basis, in order to make the closest comparison possible with an established method, in some cases we will reduce the number of electrons contained in the pseudo-potential to zero to approximate an all-electron plane-wave basis (we note that this is not the intended application of QE and we conduct this purely for validation purposes). Where we use unmodified pseudopotentials they are generated from the pslibrary (Dal Corso, 2014). We will note pseudopotential usage where relevant. FHI-aims implements numerical atomic orbital basis sets and provides the option to use periodic boundary conditions. All FHIaims calculations in this work use tight basis settings. GPAW utilizes a plane-wave basis with the pseudopotentials for which we use the recommended defaults. All calculations presented in this work use the LDA functional (Dirac, 1930; Bloch, 1929; Kohn & Sham, 1965; Perdew & Zunger, 1981).

**Results** The results of the lithium and carbon band structures are illustrated in Figure 3 and aluminium and silicon band structures are illustrated in Figure 6. The band structure indicates whether a crystal is an insulator, semiconductor, or conductor. A substantial, clear band gap suggests that the crystal primarily exhibits insulating properties. Our results, aligning consistently with existing implementations, reveal no band gap for lithium and aluminum, suggesting metallic characteristics.

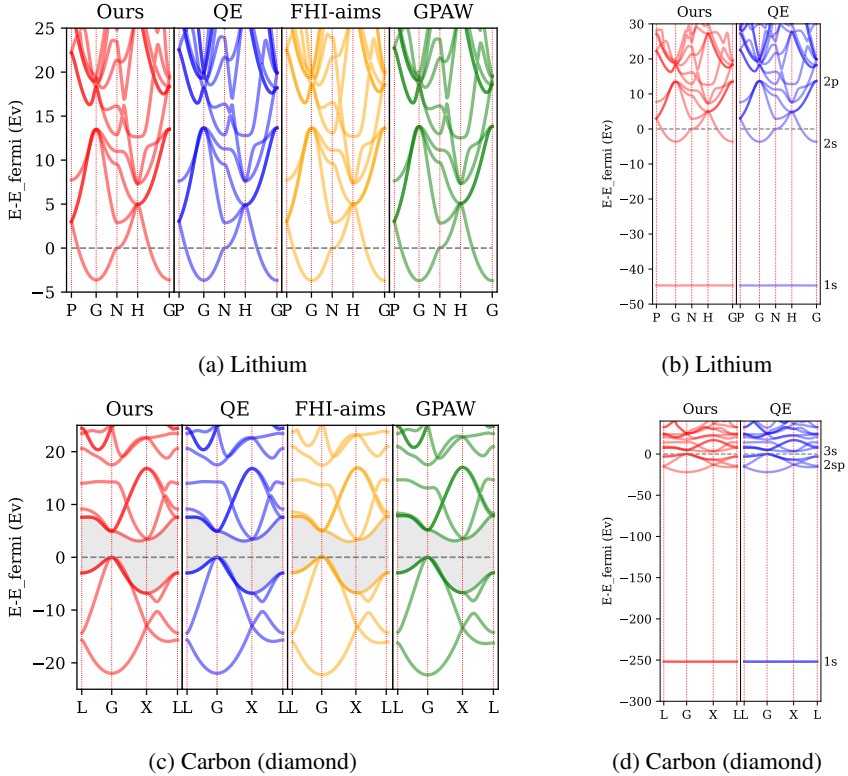

Figure 3: A Comparative illustration of band structures (part A). (a, c): a comparative analysis of the valence/conduction bands with QE, FHIaims, and GPAW, implemented on various crystals. (b, d): An exhibition of the comprehensive band structures, encompassing the core bands. In these calculations a $1 \times 1 \times 1$ $k$-point mesh is used. A cut-off energy of 200 Hartree is used for lithium, 400 for carbon for both our approach and QE, GPAW uses a cut-off energy of 400 Hartree. We adopt adam as the optimizer. QE uses an empty pseudopotential in all cases, and cold smearing (Marzari et al., 1999) with a smearing temperature of 0.01 Ry for lithium, fixed occupations for diamond.

In contrast, a significant band gap is observable for carbon, indicative of insulating properties for diamond, while a small band gap for silicon implies semiconductor behavior. This large band gap of diamond means that, at room temperature, there are no available energy states for electrons in the conduction band, and thus, no electrical current can flow through, rendering diamond an excellent insulator, whereas for metals, the valence band and the conduction band overlap, meaning there are always available energy states for electrons to move to. The conclusions drawn from our study confirm the conventional understanding of these materials and align with the theoretical predictions provided by other established implementations.

As an all-electron plane-wave method, our approach illustrates the complete band energy level spectrum of crystals, encompassing the core band. The examination of core bands is pivotal for achieving complete representation of a material's electronic structure, which, in turn, facilitates the understanding of various associated physical and chemical properties of the materials. The illustrative results of this analysis are depicted in Figure 3(b, d) and 6(b, d), with the bands names annotated based on the atomic orbital of origin on the right side for reference. The comparison reveals that our method exhibits a high degree of alignment with QE, yielding a congruent band spectrum. This congruence underscores the reliability and accuracy of our approach in analyzing and representing the intricate features of band structures, inclusive of the core bands.

## 5.2 GEOMETRY OPTIMIZATION

**Experimental setting**   We examine the structure of a primitive unit cell from a diamond-structured carbon crystal, which consists of two carbon atoms. In our experiments, one carbon atom is anchored at the origin, and we seek to ascertain the position of the second carbon atom. This atom is moving along the plane highlighted in red, as depicted in Figure 4(d). The corresponding potential energy

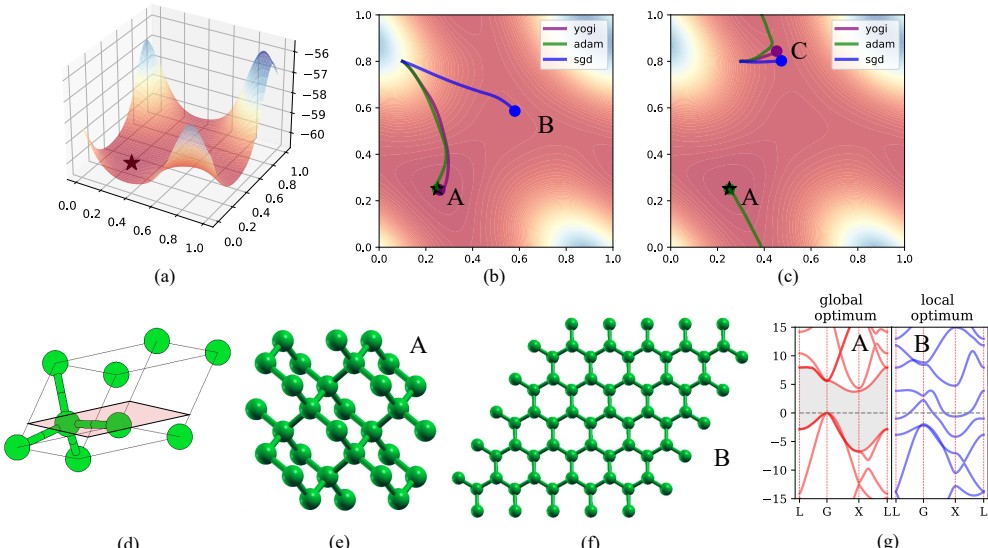

Figure 4: Illustration on the geometry optimization. The PES shown in (a-c) is computed by moving the atom in the centre of the unit-cell in (d) around the highlighted plane in a regular $50 \times 50$ grid. 'yogi', 'adam', and 'sgd' highlight optimization paths by the respective optimization method from their starting points as shown in (b, c). The global minimum is highlighted with a black star, and the geometry of the crystal at point A is given in (e), and consistent with the known structure of diamond. The geometry of the local minima at point B to which the sgd method converges to is presented in (f), which can be seen to approach the known structure of graphite. The different band structures of A and B are shown in (g). Images (d-f) generated using xcrysden (Kokalj, 2003).

surface (PES) is computed and illustrated in Figure 4(a). The optimum is denoted by the star point, accurately representing the experimentally known diamond structure. Observations reveal the presence of multiple local optima and saddle points interspersed across the PES, underscoring the complexity of the energy landscapes typically found in atomic systems.

In the experiment, a cut-off energy of 40 Hartree is employed along with a $24 \times 24 \times 24$ FFT-mesh grid, and a $2 \times 2 \times 2$ $k$-point mesh. We implement three distinct optimizers: SGD, Adam (Kingma & Ba, 2015), and Yogi (Zaheer et al., 2018), with a learning rate at $1 \times 10^{-4}$.

**Results** The results of optimizations with varied initializations are illustrated in Figure 4(b, c). In Figure 4(b), both the Yogi and Adam algorithms successfully reach the global optimum, while the SGD optimizer lands at a local minimum. Figure 4(c) demonstrates situations where both Yogi and SGD become stuck at the saddle point, whereas Adam locates the global optima in the adjacent unit cell, reflected back in the plot. This local minimum is appears to be approaching a graphite-like atomic structure, showcased in Figure 4(f). To emphasise the types of material analysis that geometry optimization can allow, we show that band structure of (e)-diamond and (f)-graphite-like in (g). In particular, there is a clear band gap in (g) A indicating insulator-like properties, while the disappearance of a band gap in (g) B implies metallic properties.

## 5.3 CONVERGENCE ANALYSIS

In this task, our attention is divided between two sub-tasks: firstly, analyzing the performance of different optimizers, and secondly, contrasting direct optimization with the SCF approach.

**Convergence of Different Optimizers.** We conducted tests using a variety of optimizers within our optimization framework, focusing on lithium and carbon crystal structures. All the optimizers employed the same initialization and a learning rate of $1 \times 10^{-5}$. Each optimizer was run ten times, with the average and 95% confidence interval displayed in Figure 5. Note that the energy we have presented is shifted and is depicted in a logarithmic scale.

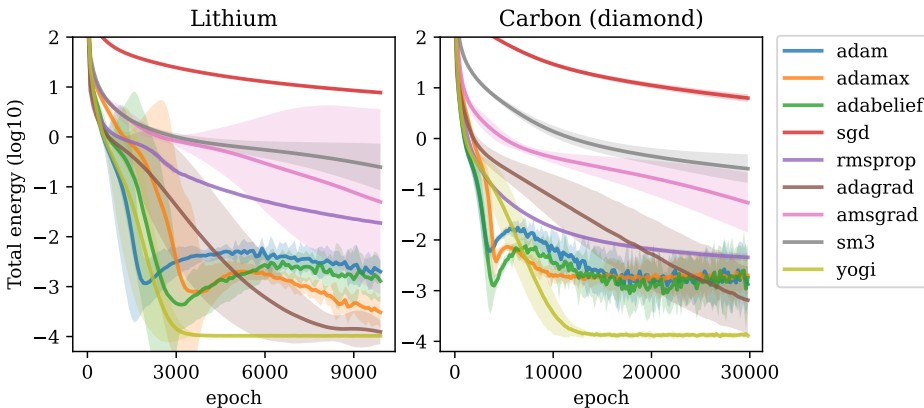

Figure 5: The convergence curves of various optimizers on lithium and carbon (diamond) crystals. The y-axis represents the logarithmic scale of the shifted total energy.

The results illustrate that Yogi consistently outperforms the other optimizers in terms of achieving lower energy, even though it does not exhibit the most rapid decrease. It is also observable that some optimizers, particularly those incorporating a momentum mechanism, are prone to experiencing bounces, manifesting as an initial faster decrease in energy, followed by an increase, before resuming the decrease. Additionally, fluctuations are more pronounced in the convergence patterns of Adam, AdaBelief, and Adamax, showcasing a higher degree of variability as they approach convergence.

**Convergence of SCF and direct optimization**   As discussed further below convergence to a stable solution using SCF methods can sometimes be challenging, something we do not expect to encounter using direct optimization. This test attempts to reproduce the PES generated in Figure 4 using static computational settings in QE and GPAW. As above a cut-off energy of 40 Hartree is used with a $48 \times 48 \times 48$ FFT-mesh, $2 \times 2 \times 2$ $k$-point mesh. QE begins all computations with an 'atomic+random' initial guess, is allowed a maximum of 300 SCF steps, and a two-electron n-kjpaw pseudopotential is used, all other settings remaining default. Where not mentioned above, GPAW calculations use program default settings.

Where we report 100% convergence of all points on the PES using our method, this test reveals 88.3% convergence for QE and 99.1% convergence for GPAW. We show the specific points the SCF calcualtions fail for in Figure 9. Note that it is expected that expert users of QE and GPAW should be capable of finding settings that allow for reliable convergence of the full PES, but that can be time consuming, and not ideal for large scale computation.

## 6   CONCLUSION

In this study, we have presented a fully differentiable method for addressing the Kohn-Sham Density Functional Theory (KS-DFT) utilizing a plane-wave basis. Unlike the conventional self-consistent field (SCF) approach, our methodology, enriched by emerging deep learning frameworks and the advancements in gradient-based optimizers, showcases robust convergence performance. We have substantiated the efficacy of our method through its application in two pivotal domains of solid-state physics: band structure prediction and geometry optimization. This methodology serves as a potential intermediary, bridging computational material science theory with the rapid advancements in deep learning infrastructures. It is our aspiration that this research will propel the pursuit of new material discoveries by leveraging computation-intensive approaches.

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

# A MORE EXPERIMENTAL RESULTS

## A.1 MORE RESULTS ON THE BAND STRUCTURE

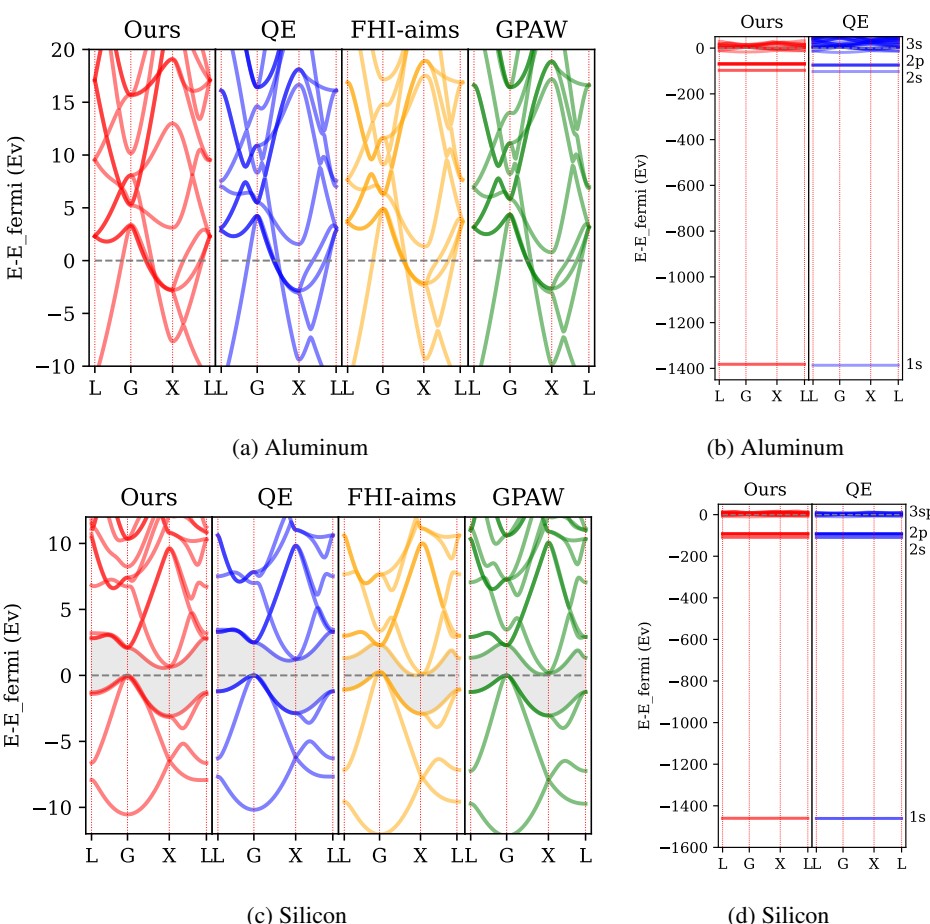

Figure 6: A Comparative illustration of band structures (part B). (a, c): a comparative analysis of the valence/conduction bands with QE, FHIaims, and GPAW, implemented on various crystals. (b, d): An exhibition of the comprehensive band structures, encompassing the core bands. In these calculations a $1 \times 1 \times 1 k$-point mesh is used. A cut-off energy of 800 Hartree for our method and QE, GPAW uses a cut-off energy of 400 Hartree. We adopt adam as the optimizer. QE uses an empty pseudopotential in all cases, and cold smearing (Marzari et al., 1999) with a smearing temperature of 0.01 Ry for aluminium, fixed occupations for silicon.

### A.2 MORE RESULTS ON THE CONVERGENCE ANALYSIS

### A.2.1 PES OF DIAMOND

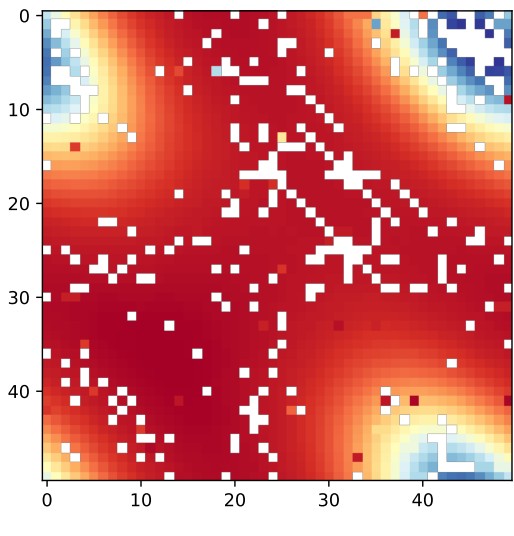

(a) SCF convergence

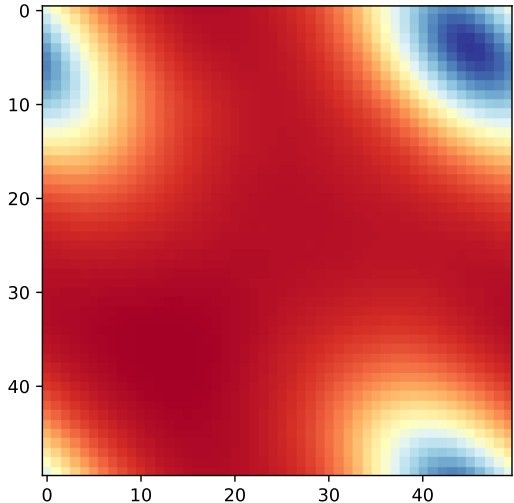

(b) Our direct optimization approach with Adam (Kingma & Ba, 2015).

Figure 7: An example of the failure of SCF in calculating $50 \times 50$ grid Potential Energy Surface (PES). A cut-off energy of 40 Hartree is used with a $48 \times 48 \times 48$ FFT-mesh, $2 \times 2 \times 2$ $k$-point mesh. QE begins all computations with an 'atomic+random' initial guess, is allowed a maximum of 300 scf steps, and a two electron n-kjpaw pseudopotential is used, all other settings remaining default. Where a calculation was unable to converge the energy is set to zero, and falls outside the scale of the colormap. The SCF convergence is conducted with Quantum ESPRESSO.

## A.2.2 PES of Lithium Fluoride

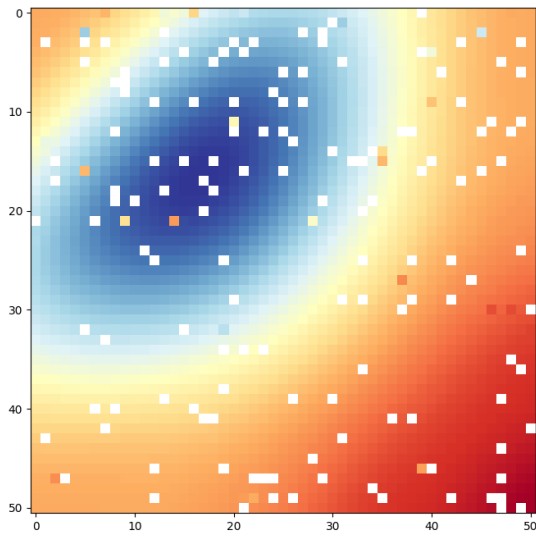

(a) SCF convergence

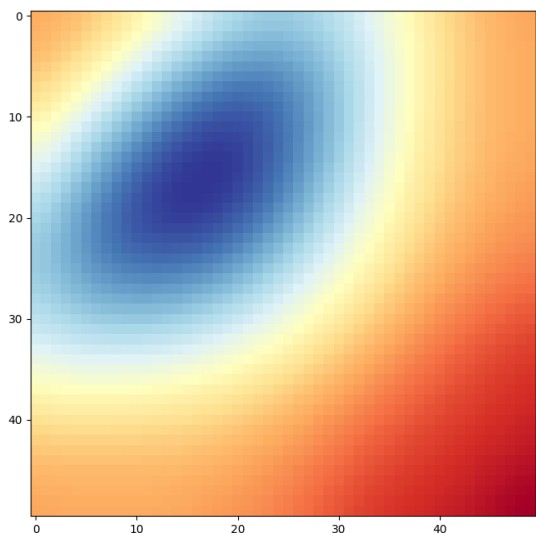

(b) Our direct optimization approach with Adam (Kingma & Ba, 2015).

Figure 8: An example of the failure of SCF in calculating $50 \times 50$ grid Potential Energy Surface (PES). A cut-off energy of 40 Hartree is used with a $1 \times 1 \times 1$ $k$-point mesh. QE begins all computations with an 'atomic+random' initial guess, is allowed a maximum of 300 scf steps, and a two electron n-kjpaw pseudopotential is used, all other settings remaining default. Where a calculation was unable to converge the energy is set to zero, and falls outside the scale of the colormap. The SCF convergence is conducted with Quantum ESPRESSO.

### A.2.3 PES OF BERYLLIUM

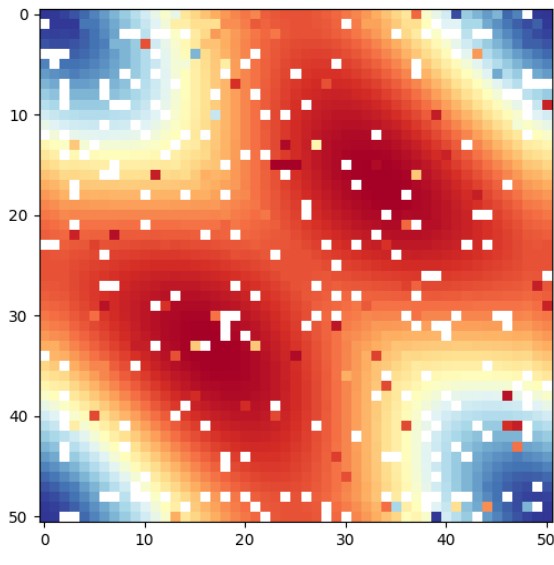

(a) SCF convergence

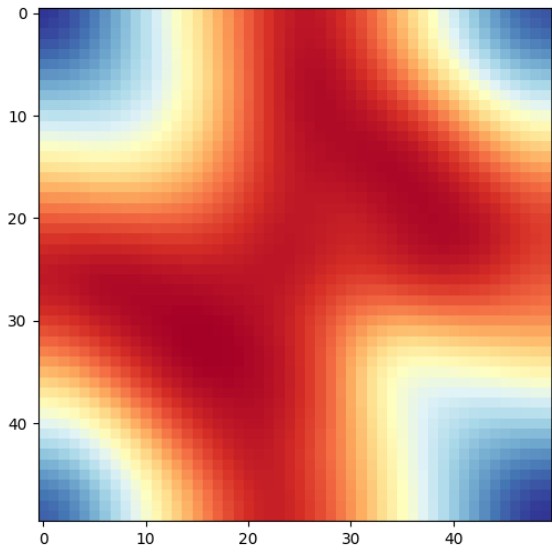

(b) Our direct optimization approach with Adam (Kingma & Ba, 2015).

Figure 9: An example of the failure of SCF in calculating $50 \times 50$ grid Potential Energy Surface (PES). A cut-off energy of 40 Hartree is used with a $1 \times 1 \times 1$ $k$-point mesh. QE begins all computations with an 'atomic+random' initial guess, is allowed a maximum of 300 scf steps, and a two electron n-kjpaw pseudopotential is used, all other settings remaining default. Where a calculation was unable to converge the energy is set to zero, and falls outside the scale of the colormap. The SCF convergence is conducted with Quantum ESPRESSO.

## B  MORE DISCUSSIONS

### B.1  RELATED WORK

**Machine learning for material science**    Numerous studies have employed deep learning techniques in the realm of material science. Some utilize supervised neural networks, leveraging data sourced from simulations or empirical observations. The prediction targets range from ground state energy (Gilmer et al., 2017), XC functional (Dick & Fernandez-Serra, 2021), (Kirkpatrick et al., 2021), (Kasim & Vinko, 2021), kinetic functional (Alghadeer et al., 2021), interatomic potential (Zeng et al., 2023), (Chmiela et al., 2017), local density of states Fiedler et al. (2023). Some are ab-initio calculations that use neural networks directly as ansatz for solving the fermionic Schrodinger equation in both atomistic and solid state settings, like Choo et al. (2020), Pfau et al. (2020), Scherbela et al. (2021), Schütt et al. (2017), Li et al. (2022), Gao & Günnemann (2022), Hermann et al. (2020), Yoshioka et al. (2021), Carleo & Troyer (2017), Pescia et al. (2022).

**Differentiable DFT for materials**    Kasim & Vinko (2021) and Zhang & Chan (2022) studied the differentiability of the Self-Consistent Field (SCF) concerning the parameters in density functional theory. Instead of traditional Lagrange multipliers, direct optimization method, introduced in (Head-Gordon & Pople, 1988), uses orthonormal basis functions (OBFs) and parameterizes the search over electronic density as unitary rotations of the OBFs, thus converting the constrained optimization to an unconstrained one. This method has undergone various enhancements, employing different parameterization techniques such as Givens rotation, Cayley transformation, and Cholesky QR factorization, cited in (Wen & Yin, 2012), (Zhang et al., 2014), and (Li et al., 2023). The adaptation of direct minimization in solid-state systems has also been significant, with approaches developed to model semiconductors and insulators, focusing on the parameterization of orthogonal planewaves coefficients and their updates, as evidenced in (Teter et al., 1989), (Štich et al., 1989), and (Kresse & Furthmüller, 1996). Extensions of these methods to metallic systems have considered finite temperature and k-point occupations, with notable advancements made by (Marzari et al., 1997), (Freysoldt et al., 2009), and (Ivanov et al., 2021), elucidating diverse methodologies in direct energy minimization and calculations of excited states.

### B.2  POTENTIAL APPLICATIONS OF THIS WORK

We outline two examples of how our work could facilitate advanced analyses and the development of novel chemical models:

- Vibrational Analysis of Systems: Our method significantly simplifies the implementation of advanced vibrational analysis techniques. Typically, vibrational analyses use the harmonic approximation, which is limited to quadratic potentials and double derivatives. However, for a more comprehensive understanding, especially in anharmonic systems, higher-order derivatives are required. The inherent capability of our approach to handle such complex derivative calculations more efficiently makes it ideally suited for extended vibrational analyses, including anharmonic vibrations. By enabling easier implementation of methods requiring triple derivatives and beyond, our work could pave the way for more accurate and detailed vibrational studies of complex materials. Other examples of properties include polarisability, dipole, quadrupole and octupole moments, Raman spectroscopy methods, and so on.

- Development of Advanced Chemical Models: Many chemical models, particularly in density functional theory (DFT), rely heavily on derivatives of properties of the chemical system. Standard density functionals typically require the density, its gradient, and Laplacian, among other derivative quantities. Our work provides a robust framework that can facilitate the testing and development of new density functionals that require novel derivatives. The ease of implementing and testing these new functionals within our framework could significantly accelerate the advancement of DFT methods and lead to more accurate models for predicting material properties.

### B.3 Differences between Molecular and plane-waves Basis

Here are some different aspects between the molecular and plane-wave basis set.

1. Assumption of System Boundaries:
    - Molecular Bases: These are typically used for finite chemical systems and utilize atom-centered basis functions, such as Gaussian type orbitals (GTOs). Implementing DFT in this context requires specifically written integral packages such as libint, tailored to these finite systems.
    - Plane-Wave Bases: In contrast, plane-wave bases assume periodic boundary conditions, allowing us to access and model technically infinite crystal systems. This requires a significantly different mathematical framing, including the definition of energy space (k-space), integration techniques, and band structure analyses.

2. Technical Implementation and Mathematical Framing: The use of a plane-wave basis necessitates a distinct approach in defining energy space, integrating over this space, analyzing band structures, and considering electron occupation across energy bands. This complexity is distinct from the challenges faced in a molecular basis.

3. Efficiency and Accuracy Considerations: While molecular calculations can approximate crystal systems (and vice versa), such approaches are generally less desirable due to efficiency and accuracy concerns. Our method, by focusing on the plane-wave basis, addresses these challenges more directly and effectively for crystal systems.

4. Specialization in Computational Chemistry Modeling: The differences between these bases are so pronounced that most computational chemistry modeling tools specialize in either a molecular basis (e.g., Q-Chem, Gaussian, ORCA) or a plane-wave basis (e.g., VASP, Quantum Espresso). This specialization underscores the distinct challenges and approaches required for each basis type.

# C  NOTATIONS

## C.1  DIRAC NOTATION

The Dirac notation (or "bra-ket" notation) is commonly used in quantum mechanics to abstractly representing quantum states and operators, without relying on a particular basis.

Quantum states are written as "ket" $|\psi\rangle$, which are column vectors of the Hilbert space of quantum states $\mathcal{H}$. The dual vectors (linear functionals) are written as "bra" $\langle\varphi|$. In finite dimensional case bras are just the conjugate transpose of $|\varphi\rangle$. The inner product between two quantum states $|\varphi\rangle$, $|\psi\rangle$ can be written as $\langle\varphi|\psi\rangle$, and similarly the outer product can be written as $|\varphi\rangle\langle\psi|$.

Once a concrete complete orthogonal basis $\{|b_i\rangle\}$ is chosen, one can express the abstract object defined above as computable expression. When the basis is discrete, like in the case of planewave expansion, the ket can be expanded as $|\psi\rangle = \sum_i \psi_i |b_i\rangle$ where $\psi_i = \langle b_i|\psi\rangle$ is the expansion coefficient. When the basis is continuous, like when dirac delta basis $|r\rangle$ is used, the summation becomes an integral: $\int \mathrm{d}r\ \psi(r)|r\rangle$, where the expansion coefficient is the function evaluation $\psi(r) = \langle r|\psi\rangle$. Correspondingly, the inner product $\langle\varphi|\psi\rangle$ can be written as

$$\langle\varphi|\psi\rangle = \sum_i \varphi_i^* \psi_i \tag{14}$$

in the discrete case, and in the continuous case we have

$$\langle\varphi|\psi\rangle = \int \mathrm{d}r\ \varphi^*(r)\psi(r). \tag{15}$$

Operators are linear maps between kets in the Hilbert space $\mathcal{H}$, analogous to matrices. Suppose an operator $\hat{O}$ maps $|\psi\rangle$ to $|\psi'\rangle$, we write $\hat{O}|\psi\rangle = |\hat{O}\psi\rangle = |\psi'\rangle$. Under a discrete basis $\{|b_i\rangle\}$ one can write out the matrix element as $O_{ij} = \langle b_i|\hat{O}|b_j\rangle$, and the linear transformation of kets can be written as a matrix-vector multiplication:

$$\psi_i' = \sum_j O_{ij}\psi_j \tag{16}$$

In the continuous case we write the matrix element as a kernel $O(r', r) = \langle r'|\hat{O}|r\rangle$, and the linear transformation becomes an integral

$$\psi'(r') = \int \mathrm{d}r\ O(r', r)\psi(r) \tag{17}$$

The expectation of observable $\hat{O}$ under wavefunction $|\psi\rangle$ is defined as $\langle\psi|\hat{O}|\psi\rangle$. Under a discrete basis it is the quadratic form

$$\langle\psi|\hat{O}|\psi\rangle = \sum_{ij} \psi_i^* O_{ij} \psi_j \tag{18}$$

and under a continuous basis it is the double integral

$$\langle\psi|\hat{O}|\psi\rangle = \int\int \mathrm{d}r'\mathrm{d}r\ \psi^*(r')O(r', r)\psi(r). \tag{19}$$

When an operator is local, like in the case of the kinetic energy operator, it can be written as $O(r', r) = O(r)\delta(r' - r)$, and the expectation becomes a single integral

$$\langle\psi|\hat{O}|\psi\rangle = \int\int \mathrm{d}r'\mathrm{d}r\ \psi^*(r')O(r)\delta(r' - r)\psi(r) = \int \mathrm{d}r\ \psi^*(r)O(r)\psi(r). \tag{20}$$

## C.2 NOTATION TABLE

We list frequently-used notations used in this paper in the next Table 1.

Table 1: Notation used in this paper

| Notation | Meaning |
| --- | --- |
| $\mathbb{R}$ | real space |
| $\mathbb{C}$ | complex space |
| $\dagger$ | Hermitian conjugate |
| $\boldsymbol{r}$ | a coordinate in $\mathbb{R}^3$ |
| $I$ | number of orbitals |
| $\psi$ | single-particle wave function / molecular orbitals |
| $u$ | periodic part of Bloch wavefunction |
| $\rho$ | electronic density |
| $\varepsilon$ | KS eigenvalue |
| $c_{i,m,n}, \boldsymbol{C_m}$ | Fourier coefficient of $u_{i,m}$ |
| $\boldsymbol{W_m}$ | learnable parameter |
| $M = M_1 \times M_2 \times M_3$ | size of the direct lattice |
| $N = N_1 \times N_2 \times N_3$ | size of the FFT-mesh |
| $\hat{H}^{\mathrm{KS}}[\rho]$ | KS Hamiltonian under density $\rho$ |
| $\boldsymbol{F_m}[\rho]$ | matrix representation of the KS Hamiltonian under density $\rho$ |
| $\boldsymbol{a}_i$ | lattice vector |
| $\boldsymbol{b}_i$ | reciprocal lattice vector |
| $\boldsymbol{m}$ | cell index |
| $\boldsymbol{R_m}$ | lattice vector |
| $\boldsymbol{k_m}$ | k point |
| $\boldsymbol{n}$ | FFT-mesh index |
| $\boldsymbol{r_n}$ | FFT-mesh lattice point |
| $\boldsymbol{G_n}$ | reciprocal FFT-mesh lattice point |
| $E_{\mathrm{el}}, E_{\mathrm{kin}}, \cdots$ | energy functionals |
| $\ell$ | atom index |
| $\boldsymbol{R}, \boldsymbol{\tau}_\ell$ | atom configuration / coordinate |
| $q_\ell$ | atom charge |
| $L$ | number of atoms in the unit cell |

# D INTRODUCTION OF KOHN-SHAM DENSITY FUNCTIONAL THEORY

## D.1 KOHN-SHAM EQUATION

In practical applications of DFT, there is a significant reliance on the Kohn-Sham equation as outlined by Kohn and Sham in (Kohn & Sham, 1965). Notably, while the original Hohenberg-Kohn formulation is based solely on electronic density, the Kohn-Sham approach is a wavefunction theory. It uses the single Slater determinant ansatz with orthonormal single-particle wave functions $\psi_i(\boldsymbol{r})$

$$\langle \psi_i | \psi_j \rangle = \delta_{ij}. \tag{21}$$

Under this ansatz the electronic density can be written as:

$$\rho(\boldsymbol{r}) = \sum_{i=1}^{I} |\psi_i(\boldsymbol{r})|^2. \tag{22}$$

where the index $I$ includes spin and electron index. The ground state density can then be obtained by minimizing the total electronic subject to the orthonormal constraint:

$$\min_{\{\psi_i\}} \quad E_{\text{el}}[\{\psi_i\}] = \sum_{i=1}^{I} \langle \psi_i | -\frac{1}{2}\nabla_{\boldsymbol{r}}^2 + \hat{V}_{\text{ext}} | \psi_i \rangle + \frac{1}{2} \int d\boldsymbol{r} \int d\boldsymbol{r}' \frac{\rho(\boldsymbol{r})\rho(\boldsymbol{r}')}{|\boldsymbol{r}-\boldsymbol{r}'|} + E_{\text{xc}}[\rho] \tag{23}$$
$$\text{s.t.} \quad \langle \psi_i | \psi_j \rangle = \delta_{ij}.$$

where the exchange-correlation (XC) energy functional $E_{\text{xc}}[\rho]$ captures the one-body exact exchange and the correlation energy. This constraint optimization problem can be solved via Lagrange multipliers. The Lagrangian is

$$\mathcal{L} = E_{\text{el}}[\{\psi_i\}] - \sum_{i,j} \lambda_{ij} \left[ \int \psi_i^*(\boldsymbol{r})\psi_j(\boldsymbol{r})d\boldsymbol{r} - \delta_{ij} \right] \tag{24}$$

where $\lambda_{ij}$ is a Hermitian $I \times I$ matrix of Lagrange multipliers. Its first order variation with respect to any $\psi_i$ for $i = 1, 2, \cdots, I$ is

$$\frac{\delta \mathcal{L}}{\delta \psi_i} = \left\{ -\frac{\nabla_{\boldsymbol{r}}^2}{2} + \underbrace{\int d\boldsymbol{r}' \frac{\rho(\boldsymbol{r}')}{|\boldsymbol{r}-\boldsymbol{r}'|}}_{:=V_{\text{H}}(\boldsymbol{r})} + V_{\text{ext}}(\boldsymbol{r}) + \underbrace{\frac{\delta E_{xc}[\rho]}{\delta \rho(\boldsymbol{r})}}_{:=V_{\text{xc}(\boldsymbol{r})}} \right\} \psi_i(\boldsymbol{r}) - \sum_{j=1}^{I} \lambda_{ij}\psi_j(\boldsymbol{r})$$
$$= \hat{H}^{\text{KS}}[\rho]\psi_i(\boldsymbol{r}) - \sum_{j=1}^{I} \lambda_{ij}\psi_j(\boldsymbol{r}) \tag{25}$$

Then, the first order condition for the Lagrangian $\frac{\delta \mathcal{L}}{\delta \psi_i} = 0$ gives the Kohn-Sham equation

$$\hat{H}^{\text{KS}}[\rho]\psi_i(\boldsymbol{r}) = \sum_{j=1}^{I} \lambda_{ij}\psi_j(\boldsymbol{r}) \tag{26}$$

Note that the Kohn-Sham Hamiltonian $\hat{H}_{\text{KS}}[\rho]$ depends on the ground state density $\rho$ due to the Hartree and XC term.

## D.2 KOHN-SHAM EIGENVALUES

Note that the solution to Eq. 26 is not unique since the Slater determinant and the density formed by the orbitals $\{\psi_i\}_i$ is the same as $\{\sum_k \psi_k U_{ki}\}_i$ where $U$ is any unitary matrix.

If we choose the $U$ that diagonalizes the Lagrange multipliers $\lambda_{ij}$, i.e. $\sum_{kl} U_{ik}\lambda_{kl}U_{lj}^\dagger = \varepsilon_i\delta_{ij}$ where the eigenvalues $\varepsilon_i$ are known as the *Kohn-Sham eigenvalues*, then we have

$$\hat{H}^{\text{KS}}[\rho] \left( \sum_k \psi_k U_{ki} \right) = \sum_j \sum_k \psi_k U_{ki}\lambda_{ij} \tag{27}$$

right multiply both side by $U^\dagger$ and we obtain the Kohn-Sham equation in the standard form

$$\hat{H}^{\text{KS}}[\rho]\left(\sum_{ki}\psi_k U_{ki}U_{il}^\dagger\right) = \sum_j \sum_{kj}\psi_k U_{ki}\lambda_{ij}U_{jl}^\dagger$$

$$\hat{H}^{\text{KS}}[\rho]\left(\sum_k \psi_k \delta_{kl}\right) = \sum_k \psi_k \varepsilon_k \delta_{kl} \tag{28}$$

$$\hat{H}^{\text{KS}}[\rho]\psi_l = \psi_l \varepsilon_l$$

From this, we see that the eigenstates of $\hat{H}^{\text{KS}}[\rho]$, also called the Kohn-Sham orbital, solve the KS Eq. 26, and any unitary rotation $U$ give rise to a particular $\lambda_{ij}$ that equivalently solves the KS Eq. 26 in the sense that it gives the same total energy.

This also implies that if the energy minimization is solved directly, we still need to diagonalize the Kohn-Sham Hamiltonian $\hat{H}^{\text{KS}}[\rho]$ to find the Kohn-Sham eigenvalues.

For a more detailed derivation of the above that considers the issue of occupation, see Lehtola et al. (2020).

### D.3 Self-consistent Field (SCF) method

The SCF method for solving the standard form of KS Eq. 28 is an iterative process

1. starts with an initial guess of KS orbitals $\{\psi_i\}$
2. calculate density $\rho$ using current guess of KS orbitals
3. diagonalize the Kohn-Sham Hamiltonian $\hat{H}^{\text{KS}}[\rho]$ at current density $\rho$ to obtain its eigenstates $\{\psi_i'\}$.
4. If $\{\psi_i'\}$ are sufficiently different from $\{\psi_i\}$, go back to step (2), otherwise return $\{\psi_i\}$.

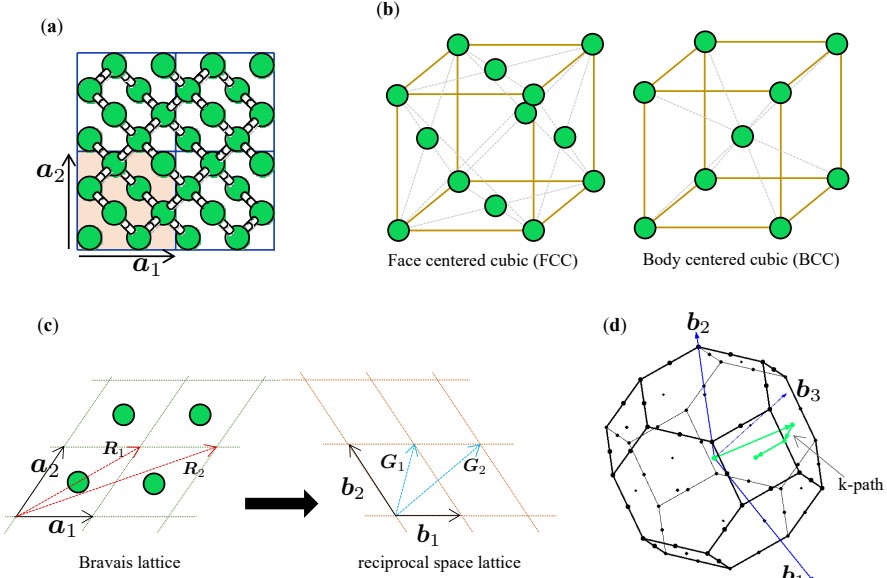

Figure 10: Review of some basic concepts of crystallography. (a) **Unit cell**: the illustration depicts a 2d diamond crystal lattice containing a grid of $2 \times 2$ unit cells. The pale orange backdrop delineates the extent of a single unit cell, with the illustration featuring two lattice vectors, denoted as $a_1$ and $a_2$. (b) **Crystal structure**: Two typical crystal structures: face-centered cubic (FCC) and body-centered cubic (BCC). (c) **Bravais and reciprocal lattices**: relationship between the two types of lattices. The transform is defined by Eq. 51. The green solid circle represent an atom in the 2d real space. The intersection points of the mesh, depicted by dotted lines, represent the $R$ vectors , whereas (d): **Brillouin zone and $k$-path sampling**: This depicts a Brillouin Zone of a face-centered cubic (FCC) structure. The black dots are the high-symmetry points. A $k$-path in the Brillouin zone is shown in a green line.

## E  INTRODUCTION TO CRYSTALLOGRAPHY

**Crystal structure**    The crystal structure of a material delineates the ordered arrangement of atoms, ions, or molecules within a crystalline solid. This arrangement bestows translational symmetry upon the material, resulting in periodic repetition throughout the atomic configuration. Various crystalline materials are known to manifest distinct crystal structures. For instance, metals such as copper and aluminum adopt a face-centered cubic (FCC) arrangement, while iron and tungsten predominantly exhibit a body-centered cubic (BCC) structure. These intrinsic crystal structures govern the distinctive properties and behaviors of different crystals, playing a pivotal role in the disciplines of materials science and solid-state physics.

**Bravais lattice**    Bravais (direct) lattices serve as a foundational framework for categorizing and detailing the periodic arrangements of atoms or molecules in crystalline materials. Typically, given the representation of a crystal lattice vector $a_1, a_2, a_3 \in \mathbb{R}^3$, the Bravais lattice of size $M_1 \times M_2 \times M_3$ is described by:

$$R_m = m_1 a_1 + m_2 a_2 + m_3 a_3, \tag{29}$$

where $m_1, m_2, m_3$ are integers and $m := (m_1, m_2, m_3)$, and $m_i = 0, 1, \cdots, M_i - 1$ for $i = 1, 2, 3$. The illustration of the lattice vector and Bravais lattice is shown in Figure 10. By considering Bravais lattice, we reduce the complexity of the massive atoms in the crystal and just focus on the fundamental repeating unit, which is usually called the **unit cell**.

**Reciprocal space** Reciprocal space is commonly used in crystallography and solid-state physics. It represents the spatial frequencies associated with the periodicity of a crystal lattice. In reciprocal space, vectors correspond to the wavevectors of plane-waves, which can represent electrons, X-rays, or other probing waves when they interact with the crystal lattice. The reciprocal lattice, a counterpart of the crystal lattice, is used to describe the diffraction patterns produced when these probing entities interact with the crystal. The mathematical definition of reciprocal lattice can by given as follows. Let's consider a real-space lattice vector $(\boldsymbol{a}_1, \boldsymbol{a}_2, \boldsymbol{a}_3)^\top$. The reciprocal lattice is then defined by

$$\boldsymbol{b}_1 = 2\pi \frac{\boldsymbol{a}_2 \times \boldsymbol{a}_3}{|\Omega_{\text{cell}}|}, \quad \boldsymbol{b}_2 = 2\pi \frac{\boldsymbol{a}_1 \times \boldsymbol{a}_3}{|\Omega_{\text{cell}}|}, \quad \boldsymbol{b}_3 = 2\pi \frac{\boldsymbol{a}_1 \times \boldsymbol{a}_2}{|\Omega_{\text{cell}}|} \tag{30}$$

where $\times$ denotes the *cross product* of two vectors, and the volume is given by $|\Omega_{\text{cell}}| = |(\boldsymbol{a}_1 \times \boldsymbol{a}_2) \cdot \boldsymbol{a}_3|$. The relationship between the real-space crystal lattice vector, and reciprocal lattice vector is that, for any $i, j$, there is

$$\boldsymbol{b}_i \boldsymbol{a}_j = 2\pi \delta_{ij}. \tag{31}$$

**FFT mesh and the Reciprocal lattice** An FFT mesh of size $N_1 \times N_2 \times N_3$ can be defined within the confines of the unit cell as:

$$\boldsymbol{r_n} = \frac{n_1}{N_1} \boldsymbol{a}_1 + \frac{n_2}{N_2} \boldsymbol{a}_2 + \frac{n_3}{N_3} \boldsymbol{a}_3. \tag{32}$$

where $n_1, n_2, n_3$ are arbitrary integers and $\boldsymbol{n} := (n_1, n_2, n_3)$, and $n_i = 0, 1, \cdots N_i - 1$ for $i = 1, 2, 3$.

The reciprocal lattice is the discrete Fourier transform of the FFT mesh, defined within the crystal's real-space unit cell. Hence it also has the size $N_1 \times N_2 \times N_3$. The reciprocal lattice vector $\boldsymbol{G}$ is defined by,

$$\boldsymbol{G_n} = n_1 \boldsymbol{b}_1 + n_2 \boldsymbol{b}_2 + n_3 \boldsymbol{b}_3, \tag{33}$$

It's worth noting that the product of a reciprocal lattice vector and a real-space lattice vector is an integer multiple of $2\pi$:

$$\boldsymbol{G_n^\top R_m} = 2\pi (n_1 m_1 + n_2 m_2 + n_3 m_3), \tag{34}$$

and similarly, for the FFT mesh we have

$$\boldsymbol{G_n^\top r_{n'}} = 2\pi \left( \frac{n_1 n_1'}{N_1} + \frac{n_2 n_2'}{N_2} + \frac{n_3 n_3'}{N_3} \right). \tag{35}$$

This relationship is deeply connected to the properties of discrete Fourier transform. When we deal with a periodic structure, its Fourier transform will yield non-zero values only at the reciprocal lattice points. This discrete nature simplifies calculations significantly.

**Brillouin zone and $k$-point sampling** The Brillouin zone is the *Wigner-Seitz primitive cell* in the reciprocal lattice of a crystalline material. It defines the range of wave vectors in a crystal's reciprocal space that provide a comprehensive representation of its electronic properties. The periodic nature of a crystal lattice leads to a periodicity in the reciprocal space as well. As a result of this periodicity, the electronic properties of a crystal within any Brillouin zone are equivalent to those in the first Brillouin zone. Thus, by focusing solely on the first Brillouin zone, one can comprehensively understand and describe the electronic properties of the crystal.

The term $\boldsymbol{k}$-points references specific locations within the Brillouin zone in the reciprocal space. Given that electron behavior in a periodic crystal spans the entire Brillouin zone, there's a requisite to sample this space to encompass all potential electron momenta. However, the combination of periodicity and symmetry negates the need to assess properties at every position within the zone. Instead, a valid chosen subset of $\boldsymbol{k}$-points provides an efficient sampling mechanism.

Over the years, various k-point sampling techniques have been developed and utilized in the realm of computational research. Notable methodologies include the Monkhorst-Pack grids Monkhorst & Pack (1976), the tetrahedron method Blöchl et al. (1994), the Chadi-Cohen method Chadi & Cohen (1973), and quasi-random sampling Umrigar & Gonze (1994). Among these, the Monkhorst-Pack method has emerged as the most predominant choice in many applications.

## F DENSITY FUNCTIONAL THEORY FOR SOLID-STATE PHYSICS

### F.1 BLOCH'S THEOREM

In this section we derive the form of electron wavefunction under periodic boundary condition (PBC) by considering the eigenfunctions of the Hamiltonian. When modeling periodic system, we usually assume a periodic potential:

$$V(\boldsymbol{r} + \boldsymbol{R_m}) = V(\boldsymbol{r}), \quad \forall \boldsymbol{R_m}. \tag{36}$$

So the full Hamiltonian is $\hat{H} = \hat{K} + \hat{V}$.

Define the translation operator $\hat{T}_{\boldsymbol{m}} = \hat{T}_{m_1,m_2,m_3}$ as

$$\hat{T}_{\boldsymbol{m}}\psi(\boldsymbol{r}) = \psi(\boldsymbol{r} + m_1\boldsymbol{a}_1 + m_2\boldsymbol{a}_2 + m_3\boldsymbol{a}_3) = \psi(\boldsymbol{r} + \boldsymbol{R_m}) \tag{37}$$

where $\boldsymbol{a}_i$ are the direct lattice vectors. Any $\hat{T}_{\boldsymbol{m}}$ commutes with the kinetic energy operator since kinetic energy is invariant when translated by lattice vectors:

$$\left[\Delta_{\boldsymbol{r}}, \hat{T}_{\boldsymbol{m}}\right]\psi(\boldsymbol{r}) = \Delta_{\boldsymbol{r}}\psi(\boldsymbol{r} + \boldsymbol{R_m}) - \hat{T}_{\boldsymbol{m}}(\Delta_{\boldsymbol{r}}\psi(\boldsymbol{r})) = 0. \tag{38}$$

The same holds for the potential operator since it is periodic:

$$\left[\hat{V}, \hat{T}_{\boldsymbol{m}}\right]\psi(\boldsymbol{r}) = V(\boldsymbol{r})\psi(\boldsymbol{r}+\boldsymbol{R_m}) - \hat{T}_{\boldsymbol{m}}(V(\boldsymbol{r})\psi(\boldsymbol{r})) = V(\boldsymbol{r}+\boldsymbol{R_m})\psi(\boldsymbol{r}+\boldsymbol{R_m}) - V(\boldsymbol{r}+\boldsymbol{R_m})\psi(\boldsymbol{r}+\boldsymbol{R_m}) = 0. \tag{39}$$

Therefore the Hamiltonian $\hat{H}$ in the periodic system commutes with all $\hat{T}_n$, which means any eigenstate $\psi$ of $\hat{H}$ is also a simultaneous eigenfunction for all $\hat{T}_n$. Now we write the eigenvalue equation for each direct lattice vector. Consider $\hat{T}_{1,0,0}$:

$$\hat{T}_{1,0,0}\psi(\boldsymbol{r}) = \psi(\boldsymbol{r} + \boldsymbol{a}_1) = e^{i2\pi k_1}\psi(\boldsymbol{r}), \quad k_1 \in [-\pi, \pi] \tag{40}$$

We can write the eigenvalue of $\hat{T}_i$ as $e^{i2\pi k_i}$ since $\hat{T}_i$ is unitary by definition. Note that this also defines the eigenvalue of $\hat{T}_{m_1,0,0}$ for any $m_1$ since

$$\hat{T}_{m_1,0,0}\psi(\boldsymbol{r}) = \underbrace{\hat{T}_{1,0,0}\dots\hat{T}_{1,0,0}}_{\times m_1}\psi(\boldsymbol{r}) = e^{i2\pi m_1 k_1}\psi(\boldsymbol{r}) \tag{41}$$

Define $k_2, k_3$ similarly. Now using the $k_j$ from these eigenstates we can create a special point in the reciprocal lattice

$$\boldsymbol{k} = k_1\boldsymbol{b}_1 + k_2\boldsymbol{b}_2 + k_3\boldsymbol{b}_3 \quad \Rightarrow \quad \boldsymbol{k} \cdot \boldsymbol{R_m} = \boldsymbol{k} \cdot \left(\sum_{i=1}^{3} m_i\boldsymbol{a}_i\right) = \sum_{i=1}^{3} 2\pi m_i k_i \tag{42}$$

This $\boldsymbol{k}$ defines the eigenvalue of any $\hat{T}_n$:

$$\psi(\boldsymbol{r} + \boldsymbol{R_m}) = \hat{T}_{\boldsymbol{m}}\psi(\boldsymbol{r}) = \hat{T}_{m_1,0,0}\hat{T}_{0,m_2,0}\hat{T}_{0,0,m_3}\psi(\boldsymbol{r}) = e^{i2\pi(\sum_{i=1}^{3} m_i k_i)}\psi(\boldsymbol{r}) = e^{i\boldsymbol{k}\cdot\boldsymbol{R_m}}\psi(\boldsymbol{r}) \tag{43}$$

Then the function $u_{\boldsymbol{k}}(\boldsymbol{r}) = e^{-i\boldsymbol{k}\cdot\boldsymbol{r}}\psi(\boldsymbol{r})$ has the same periodicity as the direct lattice: for any $\boldsymbol{R_m}$ we have

$$u_{\boldsymbol{k}}(\boldsymbol{r} + \boldsymbol{R_m}) = e^{-i\boldsymbol{k}\cdot(\boldsymbol{r}+\boldsymbol{R_m})}\psi(\boldsymbol{r} + \boldsymbol{R_m}) \tag{44}$$

$$= e^{-i\boldsymbol{k}\cdot\boldsymbol{r}}e^{-i\boldsymbol{k}\cdot\boldsymbol{R_m}}\left(e^{i\boldsymbol{k}\cdot\boldsymbol{R_m}}\psi(\boldsymbol{r})\right) \tag{45}$$

$$= u_{\boldsymbol{k}}(\boldsymbol{r}) \tag{46}$$

This means that for any $u_{\boldsymbol{k}}(\boldsymbol{r})$ periodic over the lattice, the wavefunction

$$\psi_{\boldsymbol{k}}(\boldsymbol{r}) = e^{i\boldsymbol{k}\cdot\boldsymbol{r}}u_{\boldsymbol{k}}(\boldsymbol{r}) \tag{47}$$

is an eigenstate of $\hat{H}$. The above result is the famous Bloch theorem Bloch (1929). We also have the physically interpretation of $\boldsymbol{k}$: it define the phase of eigenvalue of unit translation. Hence the number of $\boldsymbol{k}$ is naturally determined by the direct lattice dimension.

## F.2 Brillouin zone

Up to now, there is no restriction on the value allowed for $\boldsymbol{k}$. To make sure that $\psi_{\boldsymbol{k}}(\boldsymbol{r})$ respect PBC, i.e. the supercell itself is periodic, we must have

$$\psi_{\boldsymbol{k}}(\boldsymbol{r} + M_i\boldsymbol{a}_i) = \left(e^{i\boldsymbol{k}\cdot\boldsymbol{r}} \cdot e^{i\boldsymbol{k}\cdot M_i\boldsymbol{a}_i}\right) u_{\boldsymbol{k}}(\boldsymbol{r}) = \psi_{\boldsymbol{k}}(\boldsymbol{r}) \Rightarrow e^{i\boldsymbol{k}\cdot M_i\boldsymbol{a}_i} = 1 \tag{48}$$

for $i \in \{1, 2, 3\}$, where $M_i$ is the direct lattice dimension in $\boldsymbol{a}_i$ direction. Since $\boldsymbol{k} \cdot \boldsymbol{a}_i = 2\pi k_i$, we must have $k_i = \frac{m_i}{M_i}, m_i \in \mathbb{Z}$. In other words, PBC quantizes the reciprocal space into grids of size $\frac{1}{M_1} \times \frac{1}{M_2} \times \frac{1}{M_3}$.

Furthermore, $\boldsymbol{k}$ within the first Brillouin zone (FBZ), i.e.

$$k_i \in [-\lfloor (M_i - 1)/2 \rfloor, \ldots, \lfloor (M_i)/2 \rfloor] := \text{FBZ} \tag{49}$$

gives all the eigenvalues due to the degeneracy induced by periodicity in the reciprocal space: for $\boldsymbol{k} \in \text{FBZ}$, and $i, j \in \{1, 2, 3\}$ we have

$$
\begin{aligned}
&\hat{T}_i\psi_{\boldsymbol{k}+\boldsymbol{b}_j}(\boldsymbol{r}) \\
&= e^{i(\boldsymbol{k}+\boldsymbol{b}_j)\cdot(\boldsymbol{r}+\boldsymbol{a}_i)} u_{\boldsymbol{k}+\boldsymbol{b}_j}(\boldsymbol{r} + \boldsymbol{a}_i) \\
&= e^{i\boldsymbol{k}\cdot\boldsymbol{a}_i} \cdot \underbrace{e^{i\boldsymbol{b}_j\cdot\boldsymbol{a}_i}}_{=1} \cdot e^{i(\boldsymbol{k}+\boldsymbol{b}_j)\cdot\boldsymbol{r}} u_{\boldsymbol{k}+\boldsymbol{b}_j}(\boldsymbol{r}) \\
&= e^{i2\pi k_j}\psi_{\boldsymbol{k}+\boldsymbol{b}_j}(\boldsymbol{r}).
\end{aligned}
\tag{50}
$$

That is, $\psi_{\boldsymbol{k}+\boldsymbol{b}_j}$ (in general $\psi_{\boldsymbol{k}+\boldsymbol{G}}$ for any reciprocal vector $\boldsymbol{G}$) and $\psi_{\boldsymbol{k}}$ has the same eigenvalue so they are in fact the same eigenstate. In this sense the reciprocal space is really a lattice where each cell contains the same data. This makes $\boldsymbol{k}$ different from the regular linear momentum, since any $\boldsymbol{k} + \boldsymbol{G}$ represent the same state. We call $\boldsymbol{k}$ the crystal momentum.

Since given a direct lattice of size $M_1 \times M_2 \times M_3$, the number of valid $\boldsymbol{b}$ within the FBZ is finite, we can index all these k-points. Let

$$\boldsymbol{k_m} = \frac{2m_1 - M_i - 1}{2M_1}\boldsymbol{b}_1 + \frac{2m_2 - M_i - 1}{2M_2}\boldsymbol{b}_2 + \frac{2m_3 - M_i - 1}{2M_3}\boldsymbol{b}_3 \,, \tag{51}$$

where $\boldsymbol{m} := (m_1, m_2, m_3)$ are sets of three integers taking values $m_i = 1, 2, \cdots, M_i$. With this index we can write

$$\psi_{\boldsymbol{m}}(\boldsymbol{r}) = e^{i\boldsymbol{k_m}\cdot\boldsymbol{r}} u_{\boldsymbol{m}}(\boldsymbol{r}). \tag{52}$$

## F.3 K-space Decoupling via Bloch's Theorem

Under PBC, electronic density is distributed among different wavevector $\boldsymbol{k}$:

$$\rho(\boldsymbol{r}) = \sum_{\boldsymbol{m}} \rho_{\boldsymbol{m}}(\boldsymbol{r}) = \sum_{\boldsymbol{m}} \sum_{i=1}^{I} |\psi_{i,\boldsymbol{m}}(\boldsymbol{r})|^2 = \sum_{\boldsymbol{m}} \sum_{i=1}^{I} |u_{i,\boldsymbol{m}}(\boldsymbol{r})|^2. \tag{53}$$

and the orthonormal constraint becomes

$$\langle \psi_{i,\boldsymbol{m}} | \psi_{j,\boldsymbol{m}'} \rangle = \delta_{i,j}\delta_{\boldsymbol{m},\boldsymbol{m}'} \tag{54}$$

The standard form KS equation (28) is

$$\hat{H}^{\text{KS}}[\rho]\psi_{i,\boldsymbol{m}} = \psi_{i,\boldsymbol{m}}\varepsilon_{i,\boldsymbol{m}} \tag{55}$$

which is a $I \times M_1 \times M_2 \times M_3$-dimensional eigendecomposition problem. The Kohn-Sham eigenvalues here $\varepsilon_{i,\boldsymbol{m}}$ are used to produce band structure, where at $\boldsymbol{k_m}$ point of the reciprocal space, $\varepsilon_{i,\boldsymbol{m}}, i = 1, \ldots, I$ are the band values.

Since the KS orbital takes the form

$$\psi_{i,\boldsymbol{m}}(\boldsymbol{r}) = e^{i\boldsymbol{k_m}\cdot\boldsymbol{r}} u_{i,\boldsymbol{m}}(\boldsymbol{r}), \quad i = 1, \ldots, I, \tag{56}$$

due to Bloch's theorem, the KS equation can be decoupled for different k-points. Let

$$\hat{H}^{\text{KS}}_{\boldsymbol{m}}[\rho] = e^{-i\boldsymbol{k_m}\cdot\boldsymbol{r}} \cdot \hat{H}^{\text{KS}}[\rho] \cdot e^{i\boldsymbol{k_m}\cdot\boldsymbol{r}} = -\frac{(\nabla_{\mathbf{r}} + \boldsymbol{k_m})^2}{2} + V_{\text{H}}(\boldsymbol{r}) + V_{\text{ext}}(\boldsymbol{r}) + V_{\text{xc}}(\boldsymbol{r}) \tag{57}$$

Then by inserting equation 56 into equation 55 we have

$$
\begin{aligned}
\hat{H}^{\mathrm{KS}}[\rho]e^{i\boldsymbol{k_m}\cdot\boldsymbol{r}}u_{i,\boldsymbol{m}}(\boldsymbol{r}) &= e^{i\boldsymbol{k_m}\cdot\boldsymbol{r}}u_{i,\boldsymbol{m}}(\boldsymbol{r})\varepsilon_{i,\boldsymbol{m}} \\
\hat{H}^{\mathrm{KS}}_{\boldsymbol{m}}[\rho]u_{i,\boldsymbol{m}}(\boldsymbol{r}) &= u_{i,\boldsymbol{m}}(\boldsymbol{r})\varepsilon_{i,\boldsymbol{m}}
\end{aligned}
\tag{58}
$$

This allow us to solve the original problem for each $\boldsymbol{k_m}$ separately, and for each $\boldsymbol{k_m}$ we only need to solve an eigendecomposition problem of dimension $I$.

It is a common practice to approximate the ground state density $\rho$ with a coarser k-mesh $M_1 \times M_2 \times M_3$ than the one used for band structure calculation.

### F.4 Planewave Ansatz

Since $u_{i,\boldsymbol{m}}$ has the same periodicity as the direct lattice, it has discrete Fourier decomposition

$$
u_{i,\boldsymbol{m}}(\boldsymbol{r}) = \sum_{\boldsymbol{n}} c_{i,\boldsymbol{m},\boldsymbol{n}}e^{i\boldsymbol{G_n}\cdot\boldsymbol{r}}, \quad c_{i,\boldsymbol{m},\boldsymbol{n}} \in \mathbb{C},
\tag{59}
$$

so the KS orbital has form

$$
\psi_{i,\boldsymbol{m}}(\boldsymbol{r}) = e^{i\boldsymbol{k_m}\cdot\boldsymbol{r}} \cdot u_{i,\boldsymbol{m}}(\boldsymbol{r}) = \sum_{\boldsymbol{n}} c_{i,\boldsymbol{m},\boldsymbol{n}}e^{i(\boldsymbol{k_m}+\boldsymbol{G_n})\cdot\boldsymbol{r}}
\tag{60}
$$

**The orthonormal constraint** Now we consider the constraint in equation 54. The orthogonality of Bloch wave functions with distinct wave vectors $\boldsymbol{k_m} \neq \boldsymbol{k'_m}$ is ensured by the mathematical identity

$$
\langle\psi_{i,\boldsymbol{m}}|\psi_{j,\boldsymbol{m'}}\rangle = \sum_{\boldsymbol{n}} c^*_{i,\boldsymbol{m},\boldsymbol{n}}c_{j,\boldsymbol{m'},\boldsymbol{n}}\int\exp\big(\mathrm{i}(\boldsymbol{k_m}-\boldsymbol{k'_m})^{\top}\boldsymbol{r}\big)d\boldsymbol{r} = 0.
\tag{61}
$$

We only need to enforce the constraint between orbitals with the same wavevectors, i.e.

$$
\langle\psi_{i,\boldsymbol{m}}|\psi_{j,\boldsymbol{m}}\rangle = \sum_{\boldsymbol{n}} c^*_{i,\boldsymbol{m},\boldsymbol{n}}c_{j,\boldsymbol{m},\boldsymbol{n}} = \delta_{ij}, \quad \forall\boldsymbol{k_m}.
\tag{62}
$$

## G   In-Depth Introduction of the Computational Methodology.

### G.1   Mathematical fundamentals on Fourier transform

**Parseval's theorem on finite space.**   Consider a periodic function $f$ on a cell $\Omega$. The discrete Fourier transform is defined by

$$f(\boldsymbol{r}) = \frac{1}{\sqrt{|\Omega|}} \sum_{\boldsymbol{G}} \tilde{f}(\boldsymbol{G}) \exp(\mathrm{i}\boldsymbol{G}^\top \boldsymbol{r}) \tag{63}$$

$$\tilde{f}(\boldsymbol{G}) = \frac{1}{\sqrt{|\Omega|}} \int_\Omega f(\boldsymbol{r}) \exp(-\mathrm{i}\boldsymbol{G}^\top \boldsymbol{r}) \tag{64}$$

The Parseval theorem of two periodic functions $f$ and $g$ applies

$$\int_\Omega f^* g \, d\boldsymbol{r} = \sum_G \tilde{f}^*(\boldsymbol{G}) \tilde{G}(\boldsymbol{G}) \tag{65}$$

This result shows that the discrete Fourier transform maintain the inner product of two functions.

**Discrete Fourier Transform.**   Consider a sequence of points $\{a_{ijk}\}$ where $i = 0, 1, \ldots, I-1$, $j = 0, 1, \ldots, J-1$, and $k = 0, 1, \ldots, K-1$.

The 3-D discrete Fourier transform of $\{a_{\boldsymbol{r}}\}$ is defined by,

$$A_{lmn} = \sum_{i,j,k} a_{ijk} \exp\left\{-2\pi i \left(\frac{il}{I} + \frac{jm}{J} + \frac{kn}{K}\right)\right\} \tag{66}$$

and the 3D inverse FFT is defined by,

$$a_{ijk} = \sum_{l,m,n} A_{lmn} \exp\left\{2\pi i \left(\frac{li}{L} + \frac{mj}{M} + \frac{nk}{N}\right)\right\} \tag{67}$$

**The Shift Theorem**   Let $f'(\boldsymbol{r}) = f(\boldsymbol{r} + \boldsymbol{a})$, where $\boldsymbol{a}$ is a constant independent of $\boldsymbol{r}$. Then the Fourier transform of $f'$ can be written as,

$$\tilde{f}'(\boldsymbol{G}) = e^{-i\boldsymbol{G}\boldsymbol{a}} \tilde{f}(\boldsymbol{G}) \tag{68}$$

### G.2   Constructing the wave function

**Constructing the density via Fourier transform**   The density function in the reciprocal space can be written as:

$$\tilde{\rho}(\boldsymbol{G_n}) = \sum_i \sum_{\boldsymbol{m}} \sum_{\boldsymbol{n}'} c^*_{i,\boldsymbol{m},\boldsymbol{n}'} c_{i,\boldsymbol{m},\boldsymbol{n}+\boldsymbol{n}'} \tag{69}$$

The derivation of this equation can be find in the appendix. It can be seen that the Fourier transform of the density function can be expressed as the discrete convolution of coefficients. A typical way of such convolution requires $\mathcal{O}(N^2)$ steps of computation. To reduce the complexity, we implement this density function via Fourier transform.

Consider the plane-wave function defined in Eq. 59 and we evaluate it at the lattice point $\boldsymbol{r_n} = \frac{n_1}{N_1}\boldsymbol{a}_1 + \frac{n_2}{N_2}\boldsymbol{a}_2 + \frac{n_3}{N_3}\boldsymbol{a}_3$, we have,

$$u_{i,\boldsymbol{m}}(\boldsymbol{r_n}) = \sum_{\boldsymbol{n}'} c_{i,\boldsymbol{m},\boldsymbol{n}'} \exp\left(\mathrm{i}\boldsymbol{G}^\top_{\boldsymbol{n}'} \boldsymbol{r_n}\right) = \sum_{\boldsymbol{n}'} c_{i,\boldsymbol{m},\boldsymbol{n}'} \exp\left(2\pi\mathrm{i} \left(\frac{n'_1 n_1}{N'_1} + \frac{n'_2 n_2}{N'_2} + \frac{n'_3 n_3}{N'_3}\right)\right) \tag{70}$$

This equation inherently adopts the structure of inverse discrete Fourier transform applied to the sequence of coefficients, represented as:

$$u_{i,\boldsymbol{m}}(\boldsymbol{r_n}) = \mathtt{iFFT}\left(\{c_{i,\boldsymbol{m},\boldsymbol{n}'}\}\right) \tag{71}$$

Subsequently, the density function corresponding to individual KS orbitals can be calculated by

$$\rho_{i,m}(r_n) = |u_{i,m}(r_n)|^2. \tag{72}$$

The Fourier transform of the density function can be directly calculated via fast Fourier transform:

$$\tilde{\rho}_{i,m}(G_n) = \texttt{FFT}\left[\rho_{i,m}(r_n)\right] \tag{73}$$

The fast Fourier transform (Cooley & Tukey, 1965) has the complexity of $\mathcal{O}(N \log N)$, which is faster than that of the convolution operation.

### G.3 CALCULATION OF THE ENERGIES

**Total energy minimization**    In the present study, we employ the total energy minimization approach previously outlined in Li et al. (2023). Our target is to minimize the subsequent objective function:

$$
\begin{aligned}
\min_{c_{i,m,n}} \quad & E_{\text{kin}} + E_{\text{ext}} + E_{\text{har}} + E_{\text{xc}} \\
\text{s.t.} \quad & \sum_n c^*_{i,m,n} c_{j,m,n} = \delta_{ij}, \quad \forall k_m.
\end{aligned}
\tag{74}
$$

In this section we furnish the explicit expressions for each constituent component of the total energy. While we refrain from delving into the underlying derivations within the main text, a comprehensive derivation is meticulously detailed in the appendix for interested readers. The constituent energies are given by,

$$E_{\text{kin}} = \frac{1}{2} \sum_i \sum_m \sum_n c^*_{i,m,n} c_{i,m,n} \|k_m + G_n\|^2 \tag{75}$$

$$E_{\text{har}} = 2\pi \sum_{G_n \neq 0} \frac{|\tilde{\rho}(G_n)|^2}{\|G_n\|^2} \tag{76}$$

$$E_{\text{ext}} = -4\pi \sum_{G_n \neq 0} \tilde{\rho}(G_n) \sum_\ell e^{iG_n \tau_\ell} \frac{q_\ell}{\|G_n\|^2} \tag{77}$$

*Proof.* of Eq. 75

First we have,

$$\nabla^2_r \exp(iG^\top r) = \nabla^2_r \left( \cos(G^\top r) + i \sin(G^\top r) \right) \tag{78}$$

$$= \nabla_r G^\top \left( -\sin(G^\top r) + i \cos(G^\top r) \right) \tag{79}$$

$$= \|G\|^2 \left( -\cos(G^\top r) - i \sin(G^\top r) \right) \tag{80}$$

$$= -\|G\|^2 \exp(iG^\top r) \tag{81}$$

Thus the kinetic energy can be calculated in the following manner.

$$\left\langle \psi_{i,m}(r) \left| -\frac{1}{2} \nabla^2_r \right| \psi_{i,m}(r) \right\rangle \tag{82}$$

$$= \frac{1}{2} \left\langle \sum_n c_{i,m,n} \exp(i(k_m + G_n) \cdot r) \left| \sum_n \|k_m + G_n\|^2 c_{i,m,n} \exp(i(k_m + G_n) \cdot r) \right. \right\rangle \tag{83}$$

$$= \frac{1}{2} \sum_n \|k_m + G_n\|^2 c^*_{i,m,n} c_{i,m,n} \tag{84}$$

Therefore,

$$E_{\text{kin}} = \sum_i \sum_m \left\langle \psi_{i,m}(r) \left| -\frac{1}{2} \nabla^2_r \right| \psi_{i,m}(r) \right\rangle \tag{85}$$

$$= \frac{1}{2} \sum_i \sum_m \sum_n c^*_{i,m,n} c_{i,m,n} \|k_m + G_n\|^2 \tag{86}$$

$$\square$$

*Proof.* of Eq. 76

The non-interacting Coulomb potential can be written as,

$$v(\boldsymbol{r}) = \int_\Omega \frac{\rho(\boldsymbol{r}')}{\|\boldsymbol{r} - \boldsymbol{r}'\|} d\boldsymbol{r}' \tag{87}$$

Consider the Yukawa's potential,

$$v_\alpha^\dagger(\boldsymbol{r}) = \int_\Omega \frac{\rho(\boldsymbol{r}')e^{-\alpha\|\boldsymbol{r} - \boldsymbol{r}'\|}}{\|\boldsymbol{r} - \boldsymbol{r}'\|} d\boldsymbol{r}' \tag{88}$$

The Fourier transformation of the Yukawa's potential can be written as,

$$\tilde{v}_\alpha^\dagger(\boldsymbol{G}) = \int \int \frac{\rho(\boldsymbol{r}')e^{-\alpha\|\boldsymbol{r} - \boldsymbol{r}'\|}}{\|\boldsymbol{r} - \boldsymbol{r}'\|} d\boldsymbol{r}' e^{-i\boldsymbol{G}^\top \boldsymbol{r}} d\boldsymbol{r} \tag{89}$$

$$= \int \int \frac{\rho(\boldsymbol{r}')e^{-\alpha\|\boldsymbol{r} - \boldsymbol{r}'\|}}{\|\boldsymbol{r} - \boldsymbol{r}'\|} e^{-i\boldsymbol{G}^\top \boldsymbol{r}} d\boldsymbol{r} d\boldsymbol{r}' \tag{90}$$

$$(\text{shift theorem}) = \int e^{-i\boldsymbol{G}^\top \boldsymbol{r}'} \rho(\boldsymbol{r}') \int \frac{e^{-\alpha\|\boldsymbol{r}\|}}{\|\boldsymbol{r}\|} e^{-i\boldsymbol{G}^\top \boldsymbol{r}} d\boldsymbol{r} d\boldsymbol{r}' \tag{91}$$

The inner integral can be simplified using spherical coordinate (the z-axis points along the direction of $\boldsymbol{G}$):

$$\int \frac{e^{-\alpha\|\boldsymbol{r}\|}}{\|\boldsymbol{r}\|} e^{-i\boldsymbol{G}^\top \boldsymbol{r}} d\boldsymbol{r} = \int_0^{2\pi} \int_0^\pi \int_0^\infty \frac{e^{-\alpha r}}{r} e^{-i\|\boldsymbol{G}\|r\cos\theta} r^2 \sin\theta \, dr \, d\theta \, d\phi \tag{92}$$

$$= 2\pi \int_0^\pi \int_0^\infty r e^{-\alpha r} e^{-i\|\boldsymbol{G}\|r\cos\theta} \sin\theta \, dr \, d\theta \tag{93}$$

$$(u := \cos\theta) \quad = 2\pi \int_{-1}^1 \int_0^\infty r e^{-\alpha r} e^{-i\|\boldsymbol{G}\|ru} dr \, du \tag{94}$$

$$= 2\pi \int_0^\infty r e^{-\alpha r} \left[ -\frac{e^{-i\|\boldsymbol{G}\|ru}}{i\|\boldsymbol{G}\|r} \right]_{-1}^1 dr \tag{95}$$

$$= \frac{2\pi}{i\|\boldsymbol{G}\|} \int_0^\infty e^{-\alpha r} \left( e^{i\|\boldsymbol{G}\|r} - e^{-i\|\boldsymbol{G}\|r} \right) dr \tag{96}$$

$$= \frac{4\pi}{\|\boldsymbol{G}\|^2 + \alpha^2} \tag{97}$$

Therefore,

$$\tilde{v}_\alpha^\dagger(\boldsymbol{G}) = \frac{4\pi}{\|\boldsymbol{G}\|^2 + \alpha^2} \int e^{-i\boldsymbol{G}^\top \boldsymbol{r}'} \rho(\boldsymbol{r}') d\boldsymbol{r}' \tag{98}$$

$$= \frac{4\pi\tilde{\rho}(\boldsymbol{G})}{\|\boldsymbol{G}\|^2 + \alpha^2} \tag{99}$$

Further,

$$\tilde{v}_{\mathrm{H}}(\boldsymbol{G}) = \lim_{\alpha \to 0} \frac{4\pi\tilde{\rho}(\boldsymbol{G})}{\|\boldsymbol{G}\|^2 + \alpha^2} = \frac{4\pi\tilde{\rho}(\boldsymbol{G})}{\|\boldsymbol{G}\|^2} \tag{100}$$

Applying Parseval's theorem, we have

$$E_{\mathrm{H}} = \frac{1}{2} \int v(\boldsymbol{r})\rho(\boldsymbol{r}) d\boldsymbol{r} = \frac{1}{2} \sum_{\boldsymbol{G}} \tilde{v}(\boldsymbol{G})\tilde{\rho}(\boldsymbol{G}) \approx 2\pi \sum_{\boldsymbol{G} \neq 0} \frac{|\tilde{\rho}(\boldsymbol{G})|^2}{\|\boldsymbol{G}\|^2} \tag{101}$$

$\square$

*Proof.* of Eq. 110

Let $\ell$ be the index of the nuclei in the unit cell. $\tau_\ell$ is the coordinate of the $\ell$-th atom in a unit cell. The external potential can be written as,

$$v_{\text{ext}}(\boldsymbol{r}) = \frac{1}{N_{\text{cell}}} \sum_{\boldsymbol{n}}^{N_{\text{cell}}} \sum_{\ell=1}^{N_{\text{atom}}} v_\ell(\boldsymbol{r} - \boldsymbol{\tau}_\ell - \boldsymbol{R_n}) \tag{102}$$

$$\tag{103}$$

The Fourier Transformation, with the shift theorem, can be written as,

$$\tilde{v}_{\text{ext}}(\boldsymbol{G}) = \frac{1}{N_{\text{cell}}} \sum_{\boldsymbol{n}}^{N_{\text{cell}}} \sum_{\ell=1}^{N_{\text{atom}}} e^{i\boldsymbol{G}\boldsymbol{\tau}_\ell} \tilde{v}_\ell(\boldsymbol{G}) \tag{104}$$

$$:= \sum_{\ell=1}^{N_{\text{atom}}} \tilde{S}_\ell(\boldsymbol{G}) \tilde{v}_\ell(\boldsymbol{G}) \tag{105}$$

where $\tilde{S}_\ell(\boldsymbol{G})$ and $\tilde{v}_\ell(\boldsymbol{G})$ are referred to as the **structure factor** and the **form factor**, respectively, defined by

$$\tilde{S}_\ell(\boldsymbol{G}) := e^{i\boldsymbol{G}\boldsymbol{\tau}_\ell} \tag{106}$$

$$\tilde{v}_\ell(\boldsymbol{G}) := \int_{\Omega_{\text{cell}}} \frac{q_\ell}{\|\boldsymbol{r}\|} e^{-i\boldsymbol{G}\boldsymbol{r}} d\boldsymbol{r} \tag{107}$$

$$= \frac{4\pi q_\ell}{\|\boldsymbol{G}\|^2} \tag{108}$$

where $q_\ell$ is the point charge of the $\ell$-th atomic nucleus.

The overall external energy can be written as,

$$E_{\text{ext}} \approx -\sum_{\boldsymbol{G} \neq \boldsymbol{0}} \tilde{v}_{\text{ext}}(\boldsymbol{G}) \tilde{\rho}(\boldsymbol{G}) \tag{109}$$

$$= -4\pi \sum_{\boldsymbol{G_n} \neq \boldsymbol{0}} \tilde{\rho}(\boldsymbol{G_n}) \sum_\ell e^{i\boldsymbol{G_n}\boldsymbol{\tau}_\ell} \frac{q_\ell}{\|\boldsymbol{G_n}\|^2} \tag{110}$$

$$\square$$

The exchange-correlation energy is dependent on the selection of a specific functional. Potential choices for the exchange-correlation functionals encompass the local density approximation, the generalized gradient approximation, hybrid functionals, among others. Typically, the exchange-correlation energy is computed in real space utilizing numerical integration over a real-space grid, which can be written as,

$$E_{\text{xc}} = \int_{\Omega_{\text{cell}}} \varepsilon_{\text{xc}}(\boldsymbol{r}) \rho(\boldsymbol{r}) d\boldsymbol{r} \approx \frac{|\Omega_{\text{cell}}|}{N} \sum_{\boldsymbol{n}} \varepsilon_{\text{xc}}(\boldsymbol{r_n}) \rho(\boldsymbol{r_n}) \tag{111}$$

The `jax-xc` package (Zheng & Lin, 2023) is among those offering differentiable exchange-correlation functionals, which are crucial for enabling gradient-based optimization in density functional theory simulations.

Besides providing an elegant alternative to the conventional iterative solution of the KS equation and allowing easy differentiation with respect to physical parameters, our approach has a crucial advantage: it can be easily applied to the minimization of energy functionals that are given as explicit functionals of the orbitals, but not of the density. Such functionals, of which there are many examples, such as hybrids of exchange and local exchange, meta-GGAs etc., have gained traction in recent years as they tend to produce better results in critical areas such as the calculation of band gaps and the treatment of static correlation. The implementation of these functionals within the conventional Kohn-Sham scheme is an extremely demanding numerical task, which requires calculating the functional

derivatives of the orbitals with respect to the density in order to to construct the proper local effective potential. Even if this can be done, there is no guarantee that the true optimal orbitals can be generated by a local potential and therefore the resulting energy may not be a true minimum. On the other hand, our direct optimization will continue to work for these functionals with no significant increase in complexity and will continue to yield the true minimum.

# H  DETAILS FOR GEOMETRY OPTIMIZATION

## H.1  HELLMANN-FEYNMAN THEOREM

Also called the force theorem, the Hellmann-Feynman theorem Feynman (1939) describes how to take the derivative of the total energy with respect to any system parameter. Here we focus on the case of atom coordinates $\boldsymbol{R} = \{\boldsymbol{\tau}_\ell\}$. Denote the ground state total electronic energy as $\tilde{E}_{\mathrm{el}}$. For a fixed geometry $\boldsymbol{R}$, the exact ground state electronic wavefunction $\tilde{\Psi}$ is extremal with respect to any variation, so $\frac{\partial \tilde{\Psi}}{\partial \boldsymbol{R}} = 0$ and

$$\frac{\mathrm{d}\tilde{E}_{\mathrm{el}}(\boldsymbol{R})}{\mathrm{d}\boldsymbol{R}} = \left\langle \tilde{\Psi} \left| \frac{\partial \hat{H}}{\partial \boldsymbol{R}} \right| \tilde{\Psi} \right\rangle + \left\langle \frac{\partial \tilde{\Psi}}{\partial \boldsymbol{R}} \left| \hat{H} \right| \tilde{\Psi} \right\rangle + \left\langle \tilde{\Psi} \left| \hat{H} \right| \frac{\partial \tilde{\Psi}}{\partial \boldsymbol{R}} \right\rangle = \left\langle \tilde{\Psi} \left| \frac{\partial \hat{H}}{\partial \boldsymbol{R}} \right| \tilde{\Psi} \right\rangle = \int \mathrm{d}\boldsymbol{r}\, \tilde{n}(\boldsymbol{r}) \frac{\partial V_{\mathrm{ext}}(\boldsymbol{r}; \boldsymbol{R})}{\partial \boldsymbol{R}} \tag{112}$$

where $\tilde{\rho}(\boldsymbol{r})$ is the ground state density. Note that the last equality comes from the fact that the only term dependent on $\boldsymbol{R}$ in the electronic Hamiltonian is the external potential.

## H.2  EWALD SUMMATION

The main issue with evaluating the nuclear repulsion energy is that the couloumbic interaction $\frac{1}{r}$ decays slowly in real space. This means the direct lattice needs to be very large in order to correctly model it. The main idea of Ewald summation is to introduce a fast decaying function $f$, then split $\frac{1}{r}$ into a short-range term $f(\lambda r)/r$ that decays quickly in real space, and a slow-decaying term $(1 - f(\lambda r))/r$ that still decay slowly in real space but decays quickly in the Fourier space. The slow decaying can be avoided by summing the short-range term in real space and the slow-decaying term in the Fourier space. The parameter $\lambda$ controls the rate of decay.

Specifically, the decay function used in Ewald summation is the complementary error function, defined as

$$\mathrm{erfc}(r) = 1 - \mathrm{erf}(r), \quad \mathrm{erf}(r) = \frac{2}{\sqrt{\pi}} \int_0^r \mathrm{d}t\, e^{-t^2}. \tag{113}$$

The summation of the short-range term in real space is

$$E^r(\boldsymbol{R}) = \frac{1}{2} \sum_\alpha^L \sum_\beta^L \sideset{}{'}\sum_{\boldsymbol{m}} \frac{q_\alpha q_\beta \mathrm{erfc}(\lambda|\boldsymbol{\tau}_\beta - \boldsymbol{\tau}_\alpha + \boldsymbol{R_m}|)}{|\boldsymbol{\tau}_\beta - \boldsymbol{\tau}_\alpha + \boldsymbol{R_m}|} \tag{114}$$

The summation of the slow-decaying term in the Fourier space is given by (derivation omitted)

$$E^g(\boldsymbol{R}) = \frac{1}{2\Omega} \sum_\alpha^L \sum_\beta^L \sum_{\boldsymbol{G} \neq 0} q_\alpha q_\beta \frac{4\pi}{\|\boldsymbol{G}\|^2} \exp\left(-\frac{\|\boldsymbol{G}\|^2}{4\lambda^2}\right) \cdot \cos(\boldsymbol{G} \cdot (\boldsymbol{\tau}_\alpha - \boldsymbol{\tau}_\beta)) \tag{115}$$

We also need to remove the spurious self-interaction in equation 115, which is

$$E^s = \sum_\alpha^L q_\alpha \frac{\lambda}{\sqrt{\pi}}. \tag{116}$$

Thus the nuclear repulsion energy evaluated with the Ewald summation method is given by

$$E_{\mathrm{nuc}}(\boldsymbol{R}) = E^r(\boldsymbol{R}) + E^g(\boldsymbol{R}) - E^s. \tag{117}$$

