# OpenReview forum: "Differentiable Optimization in Plane-Wave Density Functional Theory for Solid States"
_ICLR.cc/2024/Conference — Submitted to ICLR 2024_

### Official Review · Reviewer_Npa9 · 2023-10-31

**Soundness:** 2 fair
**Presentation:** 3 good
**Contribution:** 2 fair
**Rating:** 5
**Confidence:** 4

**Summary:**

This manuscript follows the approach in [Li et al., 2023] to build a direct approach to solving the KSDFT equations for systems in periodic solid-state systems (by using a parameterization of the periodic part of the Bloch functions analogous to the prior work). The resulting formulation is then used for band-structure prediction and geometry optimization.

**Strengths:**

The manuscript is reasonably well written (though many details are necessarily relegated to the appendix), and does show the potential efficacy of the proposed scheme through some experiments. While, in a certain sense, the core technical insights come from [Li et al., 2023] there are invariably details to be worked out and they seem appropriately covered in detail in the appendix (though its length coupled with the reviewing timeline means I have, admittedly, not checked all the details). The problem space considered is clearly important, so the present work is certainly interesting and could have some impact.

**Weaknesses:**

Given the framing of this manuscript, a core part of the contribution is empirical illustration of the methods efficacy. (As it has to be shown to work to be interesting.) While the results are suggestive that it does, they are somewhat limited and mostly limited to qualitative comparisons with little quantitative characterization of the methods efficacy. This makes it hard to evaluate the potential impact. More concrete tests and conclusions that address these points could easily elevate the potential impact of the manuscript.

In the band structure interpolation case, no quantitative errors/discrepancies are provided for the structure. For well separated (e.g., insulating) bands this should be easy to do. Is the method accurate on a fine k-mesh? Even for "entangled" bands this could be done sufficiently far from the largest eigenvalues considered.  The plots (especially in Fig. 5) show clear differences, are these meaningful? could the accuracy of the method be improved? what is considered the "ground truth?" Similarly, often in band structure plots there are more fine-grained structures than "gap or not" that are of interest (e.g., certain types of crossings), yet those are not explored here. Absent this more detailed analysis it is hard to assess whether or not the method is effectively computing band structure.

On a related note, a common technique to go from the k-mesh to band structure on some path in k-space is Wannier interpolation. Any comparison with this is missing from the present manuscript (even if it was interpolation from the k-mesh used with the proposed method). Given the need for "fine tuning" per point on the k-mesh maybe interpolation is preferable. A comparison seems warranted. Details for this point (and the prior comments) can be found in, e.g., Section VI of [Marzari, Nicola, et al. "Maximally localized Wannier functions: Theory and applications." Reviews of Modern Physics 84.4 (2012): 1419.].

For the geometry optimization, given the ambiguity in the results having only one system makes it hard to draw conclusions. What are the conclusions beyond some optimizers going to local minima and some not in this one case? The manuscript would benefit from clarifying this point, and doing so may require more systems.

Similarly to the above points, while the single comparison with SCF for the potential energy surface is nice, it is ultimately one material and it seems pertinent to consider others across a range of "types" to be able to more strongly advocate for the relative efficacy of the method. Also, no qualitative or quantitative discussions of performance are given with classical approaches (across any experiments). While there may be advantages even if not speed, it would be good to have some sense of how such an approach stacks up with prior work (to at least better under stand the tradeoffs of using it).

**update after author response**

I would like to thank the authors for their thoughtful replies to my concerns and those of other reviewers.

In brief, the response does address some of my questions (though raises others, like just how limited is the current approach with respect to cutoff and k-point mesh—that seems important to think of it as a competing method). While I appreciate some of the small (proposed) quantitative additions to the manuscript, my overall assessment of the manuscript remains essentially unchanged.

**Questions:**

Some of the setup in Figure 2 + section 3 is slightly unclear/inconsistent and should be clarified. In the text it is suggested that the first step computes on the k-mesh (which is consistent with section 2), but the caption says "first k-point." Is this a typo? or is something else meant.

Is there any interpolation (beyond for plotting) done in Figure 2? the lower part of the figure suggests a rather sparse set of points on the path through k-space.

Additional questions specifically related to weaknesses of the manuscript are outlined in that section of the review.

---

> ### Author Response · Authors · 2023-11-17
> **Authors' response to Reviewer 4 (1/2)**
>
> **Q1 (Quantitative Comparisons)**
> > *While the results are suggestive that it does, they are somewhat limited and mostly limited to qualitative comparisons with little quantitative characterization of the methods efficacy.*
>
> > *In the band structure interpolation case, no quantitative errors/discrepancies are provided for the structure. For well separated (e.g., insulating) bands this should be easy to do. Is the method accurate on a fine k-mesh? Even for "entangled" bands this could be done sufficiently far from the largest eigenvalues considered. The plots (especially in Fig. 5) show clear differences, are these meaningful? could the accuracy of the method be improved? what is considered the "ground truth?"*
>
> Thank you for these insightful comments. We would like to address this concern by two results:
>
> - Band Gap Comparisons
>
> Our original goal for providing band structures was to demonstrate the capability of our program, however, the reviewer is correct that we have not included quantitative data and this has been an oversight. In the **table below** we compile the band gaps of diamond and silicon based on the valence band maximum and the conduction band minimum. It can clearly be seen that the band gaps of diamond are reasonably consistent, whereas the band gaps of silicon are not, which quantifies some of the error/discrepancies in bands observed by the reviewer.
>
> | Crystal | Ours | QE | FHI-aims | GPAW |
> |---|---|---|---|---|
> |diamond|3.076|3.080|2.967|3.103|
> |Silicon|0.563|1.118|0.005|0.077|
>
> The reviewer also identified that an analysis of accuracy with respect to the k-mesh and cut-off energy is absent. We note here that the k-mesh and maximum cut-off energy accessible to our method are currently modest due a need for code optimization. Consequently, calculations were conducted with a k-mesh of 1x1x1 and cut-off energy of 200/400/800 Ha as recorded in the captions of the band structures. These settings were reproduced in the other programs.
>
> The ground truth would be the computed band gap at an infinitely high k-mesh and cut-off energy which should be consistent between all programs at this limit. This is usually approximated by increasing both settings until the band gap or desired observable is converged. Consequently, the primary means to improve the accuracy of our method is to increase the k-mesh and cut-off energy, which we expect to be able to do as we optimize our method.
>
> - Energy differences of different crystal structures of carbon:
>
> Additionally, we provide a table showcasing the energy differences between points A and B from Figure 3. This measure further assesses the accuracy of our computations compared to established methods.
>
> | cutoff energy (Ha) | ours  |  QE |
> |----------------|--------|-----|
> |200 |0.18876 |	0.18860 |
> |400| 0.18753	| 0.18713 |
> | 800|0.18692| 	0.18621|
>
> These results indicate that our approach achieves comparable accuracy to conventional SCF methods, providing insight into the reliability of our computations.
>
> **Q2 (Comparison with Wannier Interpolation)**
> > *On a related note, a common technique to go from the k-mesh to band structure on some path in k-space is Wannier interpolation. Any comparison with this is missing from the present manuscript.*
>
> We acknowledge the reviewer's suggestion regarding comparing our approach with Wannier interpolation for deriving band structures along specific k-space paths. However, our manuscript's primary focus is to show the full differentiability of DFT for periodic systems, and comparing it with Wannier interpolation falls outside the intended scope of this work.
>
> We appreciate the importance of such comparisons and will duly consider them in future research endeavors that align more closely with these specific comparative analyses.

---

> > ### Author Response · Authors · 2023-11-17
> > **Authors' response to Reviewer 4 (2/2)**
> >
> > **Q3 (Geometry Optimization)**
> > > *For the geometry optimization, given the ambiguity in the results having only one system makes it hard to draw conclusions. What are the conclusions beyond some optimizers going to local minima and some not in this one case? The manuscript would benefit from clarifying this point, and doing so may require more systems.*
> >
> > Thank you for your valuable feedback on the geometry optimization section of our paper. We acknowledge the concern regarding the conclusions drawn from a single system and the need for clarity in this area. Here are some responses to this question.
> >
> > - **Purpose of the geometry optimization experiment.**
> >
> >  The primary objective of our geometry optimization experiment was to demonstrate the practical application of differentiability in the context of DFT. Our approach, as delineated in Algorithm 1, uniquely enables the joint computation of atomic geometry and plane-wave coefficients. This is distinct from conventional methods, which typically require the full convergence of the SCF loop followed by the computation of Hellmann-Feynman forces.
> >
> > - **Adaptability of optimization techniques.**
> >
> >  We intended to show the trivial adaptability of our method's optimization techniques, underscoring its flexibility and potential efficiency in different scenarios. The experiment was designed to illustrate how various optimizers could be employed in our framework and the implications of these choices for reaching local or global minima in a given system.
> >
> > - **Scope of the current work and future directions.**
> >
> >  We agree that a more extensive investigation into the most effective optimization methods for geometry optimization is warranted. However, given the vast scope of this field, as indicated by the comprehensive literature (e.g., [4] and refs contained therein), we consider such an in-depth analysis beyond the current work's scope. Geometry optimization in computational chemistry is a complex topic, encompassing hundreds of papers and various methodologies.
> >
> > In line with Reviewer 2's suggestion, we believe that a dedicated study focusing solely on optimization methods in computational chemistry would be more appropriate for a specialized venue. Our work serves as a foundational exploration, highlighting the differentiability aspect of our method and its potential applications.
> >
> > [4] Schlegel, H. Bernhard. "Geometry optimization." Wiley Interdisciplinary Reviews: Computational Molecular Science 1.5 (2011): 790-809
> >
> > **Q4 (More Comparisons)**
> > > *while the single comparison with SCF for the potential energy surface is nice, it is ultimately one material and it seems pertinent to consider others across a range of "types" to be able to more strongly advocate for the relative efficacy of the method. Also, no qualitative or quantitative discussions of performance are given with classical approaches (across any experiments).*
> >
> > As alluded to in our response to Question 2, performance comparison between this and conventional methods was not the focus of this work as further optimization and implementation of symmetry recognition is required.
> >
> > A potential energy surface (PES) of LiF and Be mimicking that presented for diamond will be included, providing a qualitative performance of SCF and our method in ionic, metallic and covalent network material types respectively. These results will be shown in the revised manuscript shortly (Appendix A).

---

> ### Comment · Reviewer_Npa9 · 2023-11-23
> **Review updated**
>
> Since I am not sure a notification is generated when the review is updated (I did not get one), this comment is to note that I have updated my review after reading the author response.

---

### Official Review · Reviewer_TgFK · 2023-11-01

**Soundness:** 4 excellent
**Presentation:** 4 excellent
**Contribution:** 3 good
**Rating:** 6
**Confidence:** 2

**Summary:**

This paper introduces a method to compute differentiable DFT solutions in the plane-wave/materials setting. This is achieved by avoiding the self-consistent equation loop where the Hamiltonian is dependent on the electron density, and the wave functions are eigenstates of the Hamiltonian. Differentiating through this SCF procedure requires differentiating through many iterations of eigendecomposition, which is highly unstable.

The proposed solution is to minimize an energy functional over a set of orthogonal vectors that has been parameterized as the QR decomposition of an unconstrained matrix. This approach requires differentiating only through one QR decomposition and the minimum of the optimization problem.

Experiments are shown where 1) band structure is determined for various materials and 2) atom positions are optimized, showcasing the differentiability of the solution.

**Strengths:**

This work addresses an important problem and opens the door to differentiating through DFT solutions, making these approaches more amenable to machine learning and optimization tools, which can have vast applications in materials science. I really like to see this problem being addressed after having run into it in a different context at one point.

The paper is well written for the conference, giving a comprehensive yet concise introduction to the problem setting.
The experiments are well chosen and cover the main ideas.

I will point out that I am not familiar with the literature landscape around these methods and will defer to the judgment of other reviewers for novelty of the method.

**Weaknesses:**

1) If I am not mistaken, there is nothing much stopping this approach from being applied to other DFT settings, such as small molecules, using some form of Gaussian-harmonic basis set. At least the orthogonality of the orbitals can be guaranteed in the same way, though one uses the orthogonality of the plane waves. If this method is capable of covering this domain as well, I wish it was done in this paper, since it would showcase the strong generality of the approach.

2) A lot of related work is mentioned, some of which close enough to allow for direct numerical comparisons. These would have improved the evaluation of the method.

**Questions:**

Would the authors be able to comment on Weaknesses point 1 and 2?

---

> ### Author Response · Authors · 2023-11-17
> **Authors' response to Reviewer 3**
>
> We would like to thank the reviewer for these insightful comments, which helped us improve the quality of this paper.
>
> **Q1 (Potential Applications)**
> >  *If I am not mistaken, there is nothing much stopping this approach from being applied to other DFT settings, such as small molecules, using some form of Gaussian-harmonic basis set.*
>
> We appreciate the reviewer's comments into the potential application of our approach to other DFT settings, such as small molecules employing Gaussian-harmonic basis sets.
>
> We would like to acknowledge that the prior work (Li et al., 2023), which is one of the most important foundation of this work, has indeed explored the extension of our methodology to diverse DFT settings, including applications involving small molecules utilizing Gaussian-harmonic basis sets. The results from this investigation supported the adaptability and effectiveness of our approach across a broader spectrum of electronic structure calculations.
>
> For a comparison of plane-wave basis and other orthogonal basis, we advise the reviewer refer to Reviewer 2 Question 2.
>
> **Q2 (More Comparisons)**
>
> > *A lot of related work is mentioned, some of which close enough to allow for direct numerical comparisons. These would have improved the evaluation of the method.*
>
> Thank you for the comments. We would like to provide some numerical comparison results against existing methods.
>
> 1. Total energy differences.
>
> We present the following tables of the difference in energy between points A and B from Figure 3. The result can measure the accuracy of our computation.
>
> | cutoff energy (Ha) | ours  |  QE |
> |----------------|--------|-----|
> |200 |0.18876 |	0.18860 |
> |400| 0.18753	| 0.18713 |
> | 800|0.18692| 	0.18621|
>
> This results shows that our approach can achieve similar results as conventional SCF methods.
>
> 2. Band gap comparison.
>
> In the table below we compile the band gaps of diamond and silicon based on the valence band maximum and the conduction band minimum. It can clearly be seen that the band gaps of diamond are reasonably consistent, whereas the band gaps of silicon are not, which quantifies some of the error/discrepancies in bands observed by the reviewer.
>
> | Crystal | Ours | QE | FHI-aims | GPAW |
> |---|---|---|---|---|
> |diamond|3.076|3.080|2.967|3.103|
> |Silicon|0.563|1.118|0.005|0.077|
>
> We note here that the k-mesh and maximum cut-off energy accessible to our method are currently modest due a need for code optimization. Consequently, calculations were conducted with a k-mesh of 1x1x1 and cut-off energy of 200/400/800 Ha as recorded in the captions of the band structures. These settings were reproduced in the other programs.

---

### Official Review · Reviewer_AFLR · 2023-11-01

**Soundness:** 3 good
**Presentation:** 3 good
**Contribution:** 3 good
**Rating:** 5
**Confidence:** 3

**Summary:**

This paper introduces a modification to density functional theory(DFT) codes which will enable back propagation through materials property predictions. The paper presents a novel and fully differentiable approach to address the Kohn-Sham Density Functional Theory (KS-DFT) in the context of solid-state physics, which is a fundamental method for studying electronic properties and structures in materials. This approach uses a plane-wave basis and is designed to leverage emerging deep learning frameworks and optimization techniques, offering robust convergence performance. The paper demonstrates the accuracy and effectiveness of the proposed approach in predicting the electronic band structures of various crystalline materials, such as lithium, aluminum, carbon (diamond), and silicon. The related works of machine learning and DFT are discussed, while a lot of details of algorithms and experiments are also present in appendix sections.

**Strengths:**

The modifications to density functional theory proposed in this paper will enable DFT to become a layer in a machine learning model, such as being used to compute the loss. Also, given how important DFT is to the problem of materials design, this improvement is bound to have an impact if it gets incorporated into DFT code bases like VASP and Quantum Espresso (there is no discussion of this in the paper).

It correctly identifies insulating, metallic, and semiconductor properties in these materials, in line with established implementations. The optimization of atomic structures within crystals are also important. The paper presents experiments showing that the proposed approach, combined with optimization algorithms like Adam and Yogi, can successfully identify optimal atomic configurations, including those for diamond and graphite-like structures. These geometry optimizations have implications for understanding the properties of materials.
The authors also compare the proposed direct optimization approach with traditional Self-Consistent Field (SCF) methods. The results indicate that the direct optimization method consistently converges, while SCF methods may encounter challenges in achieving full convergence, especially in complex energy landscapes.

**Weaknesses:**

Overall, however, I think the work would be more interesting and relevant if it had more discussion of how the work might synergize with other work, in particular materials generation and property prediction. As is, some of the impact of the paper seems left to the reader. Explicit explanation and examples of impact would be helfpul.

As such, I wonder if it would be more appropriate in a computational physics venue.

**Questions:**

Looking through the "Related Works" section, it seems like much of what was done in this paper was already done for the so-called "orthonormal basis functions", the basis used in DFT to study the chemistry of molecules, i.e. systems without a periodic crystalline structure. This paper may have just replaced these basis functions with the "plane wave" basis functions used in this paper. So it is possible this paper took established work and just changed the basis. A switch which hardly seems like a big advance, though nevertheless a useful one.

---

> ### Author Response · Authors · 2023-11-17
> **Authors' response to Reviewer 2 (1/2)**
>
> **Q1 (Potential Applications)**
>
> > *I think the work would be more interesting and relevant if it had more discussion of how the work might synergize with other work, in particular materials generation and property prediction. Explicit explanation and examples of impact would be helfpul.*
>
> Thank you for the constructive comments. We agree that a more explicit discussion on the potential synergies of our work with other areas in materials science and computational chemistry would greatly enhance its relevance and impact. Here, we outline two examples of how our work could facilitate advanced analyses and the development of novel chemical models:
>
> - Vibrational Analysis of Systems: Our method significantly simplifies the implementation of advanced vibrational analysis techniques. Typically, vibrational analyses use the harmonic approximation, which is limited to quadratic potentials and double derivatives. However, for a more comprehensive understanding, especially in anharmonic systems, higher-order derivatives are required. The inherent capability of our approach to handle such complex derivative calculations more efficiently makes it ideally suited for extended vibrational analyses, including anharmonic vibrations. By enabling easier implementation of methods requiring triple derivatives and beyond, our work could pave the way for more accurate and detailed vibrational studies of complex materials. Other examples of properties include polarisability, dipole, quadrupole and octupole moments, Raman spectroscopy methods, and so on.
>
> - Development of Advanced Chemical Models: Many chemical models, particularly in density functional theory (DFT), rely heavily on derivatives of properties of the chemical system. Standard density functionals typically require the density, its gradient, and Laplacian, among other derivative quantities. Our work provides a robust framework that can facilitate the testing and development of new density functionals that require novel derivatives. The ease of implementing and testing these new functionals within our framework could significantly accelerate the advancement of DFT methods and lead to more accurate models for predicting material properties.
>
> We will expand our discussion in the manuscript to explicitly outline these potential applications.
>
> **Q2 (Plane-wave Functions)**
>
> > *This paper may have just replaced these basis functions with the "plane wave" basis functions used in this paper. A switch which hardly seems like a big advance, though nevertheless a useful one.*
>
> We appreciate the comments and agree that differentiable DFT methods have been previously conceived in a molecular basis. However, we would like to emphasize that the implementation of these methods in a plane-wave basis, as presented in our work, is not a minor improvement but rather a significant advancement. The differences between molecular and plane-wave bases are substantial and necessitate a fundamentally different approach to the problem. Below, we detail some of these key differences:
>
> 1. Assumption of System Boundaries:
>
> - Molecular Bases: These are typically used for finite chemical systems and utilize atom-centered basis functions, such as Gaussian type orbitals (GTOs). Implementing DFT in this context requires specifically written integral packages such as libint, tailored to these finite systems.
>
> - Plane-Wave Bases: In contrast, plane-wave bases assume periodic boundary conditions, allowing us to access and model technically infinite crystal systems. This requires a significantly different mathematical framing, including the definition of energy space (k-space), integration techniques, and band structure analyses.
>
> 2. Technical Implementation and Mathematical Framing: The use of a plane-wave basis necessitates a distinct approach in defining energy space, integrating over this space, analyzing band structures, and considering electron occupation across energy bands. This complexity is distinct from the challenges faced in a molecular basis.
>
> 3. Efficiency and Accuracy Considerations: While molecular calculations can approximate crystal systems (and vice versa), such approaches are generally less desirable due to efficiency and accuracy concerns. Our method, by focusing on the plane-wave basis, addresses these challenges more directly and effectively for crystal systems.
>
> 4. Specialization in Computational Chemistry Modeling: The differences between these bases are so pronounced that most computational chemistry modeling tools specialize in either a molecular basis (e.g., Q-Chem, Gaussian, ORCA) or a plane-wave basis (e.g., VASP, Quantum Espresso). This specialization underscores the distinct challenges and approaches required for each basis type.
>
> We will ensure that our manuscript more clearly articulates these points to emphasize the significant advancements our work represents in the context of differentiable DFT methods using a plane-wave basis.

---

> ### Author Response · Authors · 2023-11-17
> **Authors' response to Reviewer 2 (2/2)**
>
> **Q3 (The Venue)**
>
> > *As such, I wonder if it would be more appropriate in a computational physics venue.*
>
> We value Reviewer 2's perspective and acknowledgment that our work aligns well with a computational physics forum. While many intended studies are indeed planned in that direction, leveraging this work as a foundation, we present this study as a novel tool for materials discovery rooted in machine learning infrastructure. Hence, we believe that ICLR serves as the appropriate venue for the debut of this work.
>
> Furthermore, the significance of materials discovery within the machine learning community is continually expanding, as evidenced by notable contributions such as works [1-3]. We believe that members of the machine learning community are well-positioned to appreciate and make effective use of the additional functionalities embedded in the method we present. Although we do plan future endeavors targeting the computational physics community, these efforts have yet to materialize.
>
>
> [1] Schütt, Kristof, et al. "Schnet: A continuous-filter convolutional neural network for modeling quantum interactions." Advances in neural information processing systems 30 (NeurIPS 2017).
>
> [2] Cho, Youngwoo, et al. "Deep-DFT: A Physics-ML Hybrid Approach to Predict Molecular Energy using Transformer." Advances in Neural Information Processing Systems (NeurIPS) Workshop. 2021.
>
> [3] Pope, Phillip, and David Jacobs. "Towards Combinatorial Generalization for Catalysts: A Kohn-Sham Charge-Density Approach."
> Advances in on Neural Information Processing Systems (NeurIPS 2023).

---

> > ### Comment · Reviewer_AFLR · 2023-11-23
> >
> > I have read the rebuttal, and believe that it has at least partially addressed my concerns. I will reconsider my rating.

---

> > > ### Author Response · Authors · 2023-11-23
> > > **Thank you for the comment**
> > >
> > > Thank you for taking the time to review our rebuttal and for your openness to reconsidering your initial assessment. We are glad to hear that we have addressed some of your concerns. If you feel that our response has positively impacted your view of our work, we would greatly appreciate it if you could update your rating to reflect your current opinion. Your revised rating would be incredibly valuable to us. Thank you again for your thoughtful consideration.

---

### Official Review · Reviewer_Arv3 · 2023-11-04

**Soundness:** 3 good
**Presentation:** 3 good
**Contribution:** 3 good
**Rating:** 3
**Confidence:** 3

**Summary:**

# Initial comment
In this paper, the authors utilize differentiable optimization to enhance the convergence robustness of solving KS-DFT, compared to the traditional SCF method. The proposed approach is implemented in JAX to utilize its automatic differentiation capability. Experiments on band structure prediction and geometry optimization are performed to show the effectiveness of the proposed method.

# After author-reviewer rebuttal
Thanks very much for your detailed response and careful work updating the manuscript. Sorry for missing the deadline to feedback to your response, as I encountered website failure.

- Your clarification has been very helpful.
- Sorry I think I made a mistake thinking you claimed efficiency in the paper.
- I find several reviewers shared the concern on the plane wave basis set selection. We don't mean it is not OK nor important. I think we are questioning the proposed method does not need to be limited to it.
- However, the biggest concern from me remains that, I am sorry to say I still find the contribution of the proposed method, compared to traditional methods and previous works, to be marginal, for both the SCF equation solving and the geometry optimization tasks.

With all the factors considered, I tend to remain my original score for now.

**Strengths:**

Using automatic-differentiation-assisted optimization instead of equation solving to improve the convergence robustness and certain optimization tasks is a promising and emerging direction.

**Weaknesses:**

The claim that the iterative nature of SCF obstructs the application of deep learning is not very suitable. See previous works such as DEQ, KSR, DQC, etc.

The experiments are not quite sufficient:
- For the band structure experiment, although several popular packages are compared, the advantages of the proposed method are not clear.
- For the geometry optimization experiment, no baseline methods are compared, while there are mature automation tools such as geomeTRIC and pyberny.

How the automatic differentiation capability of JAX is utilized is not clearly described in the paper. I will recommend Fig. 8 to be displayed in the main context with more detailed introduction.

**Questions:**

- I think the defining difference between the proposed method and the traditional method is to solve the eigenvalue problem with optimization (which has also been heard before, see https://neurips.cc/virtual/2023/poster/70089). But then to me, this is still an iterative method. So can the authors explain why in the paper it is emphasized to overcome the iterative nature of the traditional method?
- Besides the robustness of convergence, efficiency is also claimed in the paper. However, it seems that the claimed advantage in efficiency is not demonstrated?
- For the geometry optimization task, traditionally there can be analytical derivatives available. So I wonder if there is any advantage of using the proposed method?
- Can the authors further clarify why is the plane-wave basis set emphasized?

---

> ### Author Response · Authors · 2023-11-17
> **Authors' response to Reviewer 1 (1/2)**
>
> We would like to thank the reviewer for these insightful comments, which helped us improve the quality of this paper.
>
> **Q1 (The Advantages of Direct Optimization over SCF)**
> > *The claim that the iterative nature of SCF obstructs the application of deep learning is not very suitable.*
>
> We appreciate the reviewer's perspective regarding the iterative nature of gradient-based optimization and its potential compatibility with deep learning methods.
>
> We want to emphasize that SCF methods can fail to find solutions in some cases (for example near-degeneracies or highly correlated electrons [1]). Direct optimization in our method can avoid these failures.
>
> We also wish to emphasize that the differentiability of our method provides more mechanisms to interface with deep learning architectures than existing SCF methods. For example, the differentiable KS-DFT method proposed in [2] involves a scf loop in the computational graph, where the gradient backpropagation goes through all the scf iterations and which requires a fully converged solution to the scf loop and risks the failure mentioned above. In contrast, the differentiable optimization enables the development of end-to-end trainable models without going back and forth between SCF in another package, and neural networks in deep learning framework.
>
> We apologize for the poor phrasing, and have modified our manuscript to make these points more clear.
>
>
> [1] Schlegel, H. Bernhard, and J. J. W. McDouall. "Do you have SCF stability and convergence problems?." Computational advances in organic chemistry: Molecular structure and reactivity (1991): 167-185.
> [2] Kasim, Muhammad F., and Sam M. Vinko. "Learning the exchange-correlation functional from nature with fully differentiable density functional theory." Physical Review Letters 127.12 (2021): 126403.
>
> **Q2 (Band Structure)**
> >  *For the band structure experiment, although several popular packages are compared, the advantages of the proposed method are not clear.*
>
> Thank you for this comment. We would like to emphasize that the band structure experiments are a standard application for solid-state DFT computation. The purpose of this experiment is to show the efficacy of our direct optimization approach, which can produce comparable results to SCF methods. Here we are not claiming that we outperformed these popular packages. However, there are some potential advantages that our method might possess due to its full differentiability nature. For instance, we should be able to design pseudo potentials or exchange-correlation functionals with guidance of the band gap, which is more easy to access from experiments and no existing methods have done so, to the best of our knowledge. This is an intriguing application that we will explore further.
>
> For more quantitative comparison of band structure, please refer to Reviewer 4 Question 2.
>
> **Q3 (Geometry Optimization)**
>
> >  *For the geometry optimization experiment, no baseline methods are compared, while there are mature automation tools such as geomeTRIC and pyberny. …traditionally there can be analytical derivatives available. So I wonder if there is any advantage of using the proposed method?*
>
> Thank you for your insightful comments and we acknowledge the importance and widespread use of analytical derivatives in traditional SCF methods for geometry optimization.
>
> The key advantage of our approach is its inherent adaptability and flexibility. By integrating the computation of derivatives directly into the optimization process, our method simplifies the workflow and reduces the complexity of implementing geometry optimization, especially in systems where deriving analytical expressions is challenging or impractical.
>
> Furthermore, our method's capacity to handle derivative calculations internally lends itself to a broader range of applications. This includes complex systems where traditional methods might struggle due to the intricate nature of deriving and implementing analytical derivatives. As detailed in our response to Reviewer 2 Question 1, we foresee numerous potential applications in future work, such as handling complex molecular systems, materials with unconventional properties, and scenarios where rapid prototyping of computational models is essential.
>
> In light of your feedback, we recognize the necessity to clarify these points in our manuscript. We will include a more detailed discussion of the advantages of our method compared to traditional approaches, backed by quantitative analyses where applicable.

---

> > ### Author Response · Authors · 2023-11-17
> > **Authors' response to Reviewer 1 (2/2)**
> >
> > **Q4 (Automatic Differentiation)**
> > > *How the automatic differentiation capability of JAX is utilized is not clearly described in the paper. I will recommend Fig. 8 to be displayed in the main context with more detailed introduction.*
> >
> > We appreciate this point. The utilization of JAX's automatic differentiation forms a foundational aspect of our methodology, enabling seamless integration of differentiable optimization into electronic structure calculations (as highlighted in our response to Q1). We will revise the paper to provide a more comprehensive and explicit description of how JAX's automatic differentiation features are harnessed in our approach. Specifically, we will elaborate on how these functionalities empower our method in optimizing the electronic structure while ensuring computational efficiency and accuracy.
> >
> > Regarding Fig. 8, we acknowledge its significance in showcasing key aspects of our methodology. We will incorporate this figure into the main body of the paper and provide a detailed introduction that elucidates the specific components and their role in our approach.
> >
> > **Q5 (Efficiency)**
> > > *Besides the robustness of convergence, efficiency is also claimed in the paper. However, it seems that the claimed advantage in efficiency is not demonstrated?*
> >
> > Thank you for your comment regarding the efficiency claims in our paper.  Upon reviewing our manuscript, we confirm that our primary emphasis has been on the robustness of convergence and the accessibility of differentiable methods in the context of plane-wave density functional theory. While we discuss efficiency in terms of these aspects, we have NOT explicitly claimed that our method is more efficient in a general computational sense compared to traditional methods.
> >
> > However, we recognize the importance of demonstrating and quantifying efficiency in computational physics. In this regard, we believe our method shows potential advantages in specific areas:
> >
> > - Reliability in Finding SCF Solutions: One of the notable efficiencies of our approach is its reliability in achieving SCF solutions, particularly in systems where traditional methods may struggle. This reliability can be viewed as an efficiency gain, as it potentially reduces the time and computational resources required to reach a viable solution.
> >
> > - Ongoing Optimization of Implementation: We are actively working on optimizing the implementation of our method. Future versions are expected to show improvements in computational speed and resource utilization, which we plan to comprehensively benchmark against established methods.
> >
> > - Advantage in Differentiable Methods: The access to differentiable methods within our framework presents an efficiency in terms of workflow and application versatility. It enables a more direct and streamlined approach to solving complex problems, which can be particularly efficient in research areas where differentiability is a key requirement.
> >
> > We acknowledge the need to provide a more explicit discussion on these points,  and we will make necessary modifications in our manuscript.
> >
> > **Q6 (The Plane-wave Basis)**
> >
> > > *Can the authors further clarify why is the plane-wave basis set emphasized?*
> >
> > We appreciate the reviewer's comments regarding the emphasis on the plane-wave basis set in our manuscript. The choice and emphasis on the plane-wave basis set stem from its widespread usage and several advantageous properties that make it particularly suitable for PERIODIC electronic structure calculations.
> >
> > The plane-wave basis set offers unique benefits, notably its ability to efficiently represent electronic wavefunctions in reciprocal space. This characteristic is crucial for accurately describing the periodic nature of crystalline solids, making it a preferred choice for computations involving periodic systems.
> >
> > Moreover, the plane-wave basis set exhibits rapid convergence properties with respect to the kinetic energy cutoff, allowing for systematic improvements in accuracy by increasing the basis set's energy cutoff. This convergence behavior is advantageous for achieving high precision without an excessive increase in computational cost.
> >
> > Additionally, the plane-wave basis set facilitates straightforward implementations in various electronic structure methods, enabling seamless integration with diverse algorithms and techniques for studying material properties.
> >
> > By emphasizing the plane-wave basis set in our work, we aim to highlight its fundamenal role as a robust and versatile tool in electronic structure calculations, particularly in the context of our proposed methodology. We acknowledge the importance of explicitly elucidating the rationale behind this emphasis and will revise the manuscript to provide a more detailed and comprehensive explanation of the unique advantages offered by the plane-wave basis set.
> >
> > For a comparison of plane-wave basis and other orthogonal basis, we advise the reviewer refer to Reviewer 2 Question 2.

---

### Author Response · Authors · 2023-11-20
**Looking forward to further discussion**

Dear Reviewers,

We greatly appreciate your thorough and insightful feedback, which has been instrumental in enhancing our manuscript. We've submitted a revised version, incorporating responses to all reviewer comments. We are confident that this updated manuscript addresses the key issues raised and are open to making further modifications if needed.

We recognize the demanding nature of this period and the significant time and effort required for paper reviews. With the discussion deadline nearing, we look forward to any additional comments you may have on our revised manuscript and our responses to your initial feedback. Thank you for your active participation in this review process, and we welcome any final queries or concerns you might have.

Warm regards,

The Authors

---

### Meta-Review · Area_Chair_KEYk · 2023-12-05

**Metareview:**

This paper introduces a differentiable Kohn-Sham density functional theory (DFT) solver in a plane-wave basis for solid-state systems. It achieves this by using a parameterization of the periodic part of the Bloch functions. The resulting formulation is then employed for band-structure prediction and geometry optimization.

The reviewers found the paper to be reasonably well-written. However, they noted a lack of numerical evidence convincingly demonstrating the efficacy of the method, particularly when compared to classical numerical methods. The reviewers generally agreed that any observed advantages over classical methods were not clear, particularly when considering cut-off and k-point discretization. They also pointed out the narrow scope of the benchmarking for geometry optimization.

While the authors addressed several of the reviewers’ comments, their responses also raised further issues.

Given the unclear advantages of the proposed method over classical methods, and the lack of evidence supporting its purported downstream benefits, I recommend rejection.

**Justification For Why Not Higher Score:**

The comparison of the method versus classical methods is not complete, and lackluster, undermining the claim of efficacy. There are no enough evidence supporting the claims of benefits for downstream tasks.

**Justification For Why Not Lower Score:**

N/A

---

### Decision · Program_Chairs · 2024-01-16

Reject